# A 3D printable alloy designed for extreme environments

Timothy M. Smith[1 ✉], Christopher A. Kantzos[1], Nikolai A. Zarkevich[2], Bryan J. Harder[1], Milan Heczko[3], Paul R. Gradl[4], Aaron C. Thompson[5], Michael J. Mills[3], Timothy P. Gabb[1] & John W. Lawson[2]

Multiprincipal-element alloys are an enabling class of materials owing to their impressive mechanical and oxidation-resistant properties, especially in extreme environments[1,2]. Here we develop a new oxide-dispersion-strengthened NiCoCr-based alloy using a model-driven alloy design approach and laser-based additive manufacturing. This oxide-dispersion-strengthened alloy, called GRX-810, uses laser powder bed fusion to disperse nanoscale $Y_2O_3$ particles throughout the microstructure without the use of resource-intensive processing steps such as mechanical or in situ alloying[3,4]. We show the successful incorporation and dispersion of nanoscale oxides throughout the GRX-810 build volume via high-resolution characterization of its microstructure. The mechanical results of GRX-810 show a twofold improvement in strength, over 1,000-fold better creep performance and twofold improvement in oxidation resistance compared with the traditional polycrystalline wrought Ni-based alloys used extensively in additive manufacturing at 1,093 °C[5,6]. The success of this alloy highlights how model-driven alloy designs can provide superior compositions using far fewer resources compared with the 'trial-and-error' methods of the past. These results showcase how future alloy development that leverages dispersion strengthening combined with additive manufacturing processing can accelerate the discovery of revolutionary materials.

High-entropy alloys, also commonly referred to as multi-principal element alloys (MPEAs), are a class of materials that are currently of interest among the metallurgical community[1,2,7–9]. In the past decade numerous scientific investigations have uncovered remarkable properties exhibited by these alloys[7,10–13]. One of the most heavily investigated MPEA family is the Cantor alloy CoCrFeMnNi and its derivatives[2,8,14]. This group of alloys showed excellent strain hardening, resulting in high tensile strength and ductility[7,15–18]. Overcoming the strength–ductility trade-off is a result of atomic-scale deformation mechanisms[16], such as locally variable stacking-fault energies[19] and magnetically driven phase transformations[20]. This class of alloys has also proven to be robust, resisting hydrogen environment embrittlement[21], exhibiting improved irradiation properties[22] and providing superior strength at cryogenic temperatures[23]. As a result, these alloys show great potential for numerous aerospace and energy applications in elevated-temperature and corrosive environments, allowing for weight reduction and higher performance operation.

One Cantor alloy derivative of special interest is the medium-entropy alloy NiCoCr. This alloy family provides the highest strength at room temperature among the Cantor alloy and its derivatives[2,24]. Recently, this alloy was shown to provide impressive tensile properties (1,100 MPa room temperature yield strength) when undergoing partial recrystallization heat treatment after cold rolling[17]. These properties are also attributed to strain-induced, face-centred cubic (FCC) to hexagonal

close-packed (HCP) phase transformations and local stacking-fault variations. Alloying and doping of NiCoCr with refractory elements and interstitials have also been explored recently. Seol et al. found that doping the high-entropy alloy, NiCoCrFeMn, with 30 ppm of boron resulted in significant improvements in strength and ductility attributed to both grain boundary and interstitial strengthening from the boron[25]. Recent studies have also found that the addition of carbon to MPEAs resulted in improved strength[26–28]. Lastly, Wu et al.[29] found that three atomic percentage (at.%) additions of W in NiCoCr created a finer grain structure (average grain size 1 μm), resulting in a large increase in yield strength of the alloy (over 1,000 MPa, compared with 500 MPa for nonalloyed NiCoCr) while maintaining exceptional ductility of over 50% (ref. 29). These results suggest that significant improvements in FCC MPEA systems can still be realized through additional alloying.

Investigations of oxide-dispersion-strengthened (ODS) MPEAs have shown improved high-temperature properties (strength and creep)[4] and irradiation properties[30]. Similarly, multiple recent studies have successfully produced ODS alloys through laser powder bed fusion (L-PBF) using a variety of techniques[3,4,31]. These methods have relied on mechanical alloying[4,31], in situ alloying[3] or chemical reactions[32] to introduce and incorporate oxides into the three-dimensional (3D) printed matrix. However, all these processes introduce complexity and repeatability issues when trying to produce similar material through

[1]NASA Glenn Research Center, Cleveland, OH, USA. [2]NASA Ames Research Center, Moffett Field, CA, USA. [3]Department of Materials Science and Engineering, The Ohio State University, Columbus, OH, USA. [4]Propulsion Department, NASA Marshall Space Flight Center, Huntsville, AL, USA. [5]HX5 LLC, Fort Walton Beach, FL, USA. ✉e-mail: timothy.m.smith@nasa.gov

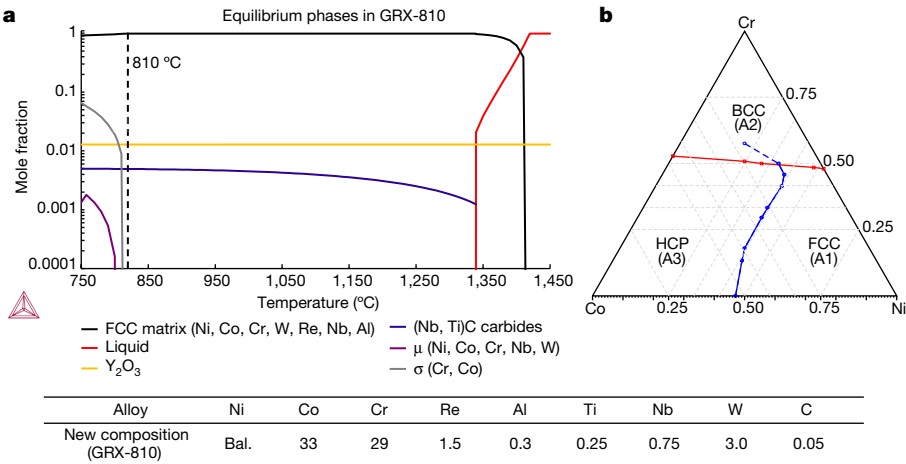

**Fig. 1 | Modelling of GRX-810 and the NiCoCr compositional space.**
**a**, Predicted phase stability in GRX-810. **b**, Calculated NiCoCr ternary phase diagram at 0 K. **a**, The body-centred cubic (BCC) Cr-rich phase has lower energy than FCC or HCP above the red line; dashed blue line indicates $E$(HCP) = $E$(FCC)

for metastable HCP and FCC, which are higher in energy than BCC. Solid blue line separates the lowest energy HCP Co-rich and FCC Ni-rich phases. Points are assessed from density functional theory (DFT) as shown in Extended Data Fig. 3. Values are at.%. Table shows nominal composition of GRX-810 (in wt%).

different additive manufacturing (AM) methods or machines. Recent work by Smith et al. produced ODS NiCoCr through L-PBF in which nanoscale $Y_2O_3$ nanoparticles were coated onto NiCoCr metal powder through a high-energy mixing process that does not require any binders, fluids or chemical reactions. This process did not deform or impact powder spherical morphology, which is important in regard to high-quality AM components. Using this approach these authors produced an ODS alloy that provided a 35% increase in tensile strength and threefold improvement in ductility at 1,093 °C compared with its non-ODS counterpart[33].

Building on the work and using the same coating process performed by Smith et al.[33], a model-driven alloy design approach was employed to optimize the NiCoCr alloy system for high-temperature applications using AM for complex components. This effort resulted in a new composition that was built using L-PBF to include nanoscale $Y_2O_3$ dispersoids for further high-temperature strength/stability above 810 °C. The characterization of this new alloy, Glenn Research Center Extreme Temperature above 810 °C (GRX-810), showed orders of magnitude better creep strength and twofold higher tensile strength compared with commercially available high-temperature alloys used in AM[34,35] and with other alloys explored in this study (NiCoCr, NiCoCr-ODS, NiCoCr-ODS with minor additions of Re (1.5 wt%) and B (0.03wt%) (ODS-ReB)). This study confirms the maturity of both model-driven alloy design and AM processes to produce next-generation materials with properties not feasible through previous, conventional manufacturing technologies.

## Microstructure characterization of GRX-810

Figure 1 provides the predicted phase equilibrium of the model-optimized GRX-810 alloy and its composition, based on percentage by weight. A full description of the modelling approach and microstructure analysis of GRX-810 can be found in Methods. The phase diagram in Fig. 1b shows that, for a large segment of the NiCoCr compositional space, HCP is the most energetically stable phase at 0 K. However, due to the higher symmetry and entropy associated with the FCC phase, this is expected to be observed at elevated temperatures, as reported for decades in NiCoCr-based alloys[36–39]. Figure 2 provides the high-resolution microstructural characterization of untested hot isostatic pressurized (HIP) GRX-810 after the powder was coated and consolidated through AM as shown in Extended Data Figs. 1 and 2.

One notable observation in Fig. 2a is the presence of carbon segregation along some, but not all, oxide–matrix interfaces. The additional

low-angle annular dark-field (LAADF)–STEM DCI analysis shown in Fig. 2b shows a representative defect microstructure. It consists of network of 1/2<110> dislocations mostly dissociated into observable intrinsic stacking faults bound by 1/6<112> Shockley partials. Dissociated dislocations mutually interact and form numerous extended stacking-fault node configurations. The density of these dissociated dislocations and grain structure of GRX-810 is better shown in the lower resolution microstructural characterization shown in Extended data Fig. 4. In addition, the presence of numerous stacking-fault tetrahedra and prevalent dislocation interaction with oxides is observed. Stacking-fault tetrahedra have been found to further inhibit dislocation motion and may further improve the creep and tensile properties of this alloy[40]. Figure 2c,d shows solute segregation of Cr, W and Re at the grain boundary, with Ni and Co depleted. The EDS map in Fig. 2c also shows the presence of Nb/Ti-rich metal carbides predicted by the thermodynamic models to be stable up to alloy melting temperature. This analysis was further validated through SEM as shown in Extended Data Fig. 5. High-resolution high-angle annular dark-field (HAADF)–STEM analysis of the GRX-810 lattice was performed to explore whether local chemical ordering exists in this alloy, as has been found in other high-entropy alloys[41,42]. The analysis in Fig. 2e,f shows that, despite possessing $L1_2$-forming elements such as Al, Ti and Nb, the lattice maintained a perfect solid solution with no short-range elemental ordering present[43].

## Mechanical properties of GRX-810

Five different MPEA alloys (NiCoCr, NiCoCr-ODS, ODS-ReB, GRX-810 and non-ODS GRX-810) under both as-built and HIP conditions were tensile and/or creep tested at 1,093 °C to compare their overall high-temperature mechanical properties. Tests were also performed on AM 718, AM 625 and wrought Haynes 230 for comparison with conventional wrought superalloys used extensively in AM. Figure 3 shows the tensile and 20 MPa creep performance of these alloys at 1,093 °C.

Figure 3a shows the elevated-temperature tensile tests (1,093 °C) that highlight the strength and elongation differences of the five alloys tested. The non-ODS NiCoCr sample was found to have lower strength and ductility than the NiCoCr-ODS sample. In fact, by simply incorporating $Y_2O_3$ particles the strength of NiCoCr was increased and ductility improved twofold. This highlights the strengthening effect provided by these oxides at elevated temperatures. The minor additions of Re

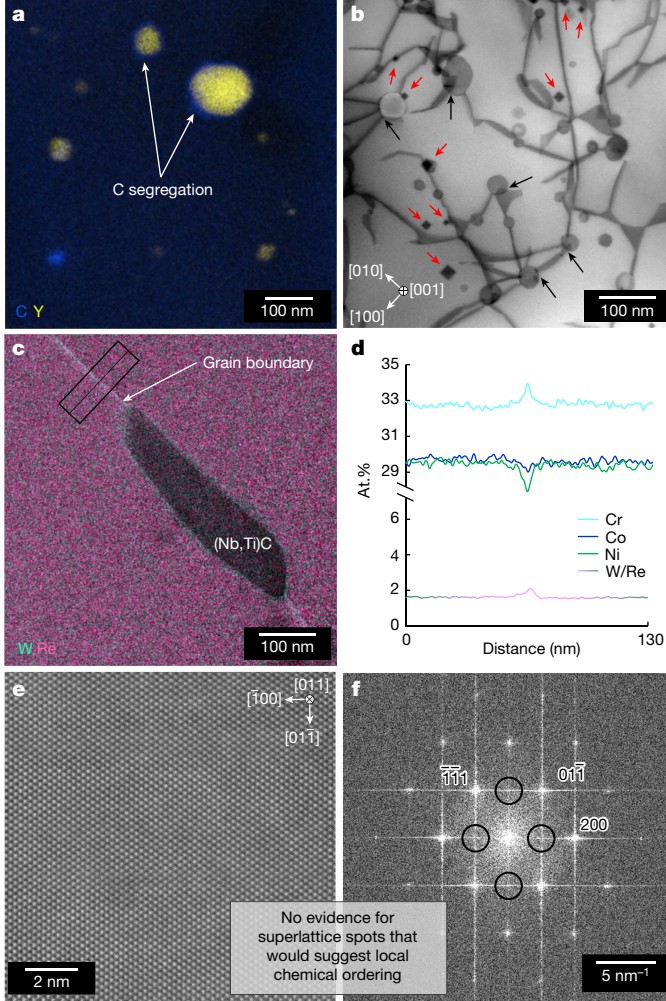

**Fig. 2 | High-resolution characterization of GRX-810 microstructure.**
**a**, Scanning transmission electron microscopy–energy dispersive X-ray
spectroscopy (STEM–EDS) combined Y and C map showing C segregation at the
oxide–matrix interface. **b**, BF–STEM diffraction contrast image (DCI) micrograph
(electron beam is parallel with [001] zone axis of matrix) of dislocation
interaction with oxides (black arrows) and the presence of stacking-fault
tetrahedra (red arrows). **c**, STEM–EDS combined W and Re map showing
segregation at grain boundary and surrounding the carbide. **d**, Integrated line
scans (at.%), from the rectangle outlined in **c**, showing segregation of Cr, W and
Re and depletion of Co and Ni at the grain boundary. Elements not measuring
any change across the boundary are not shown. **e**, Atomic resolution [011] zone
axis HAADF–STEM image of GRX-810 lattice. **f**, Fast Fourier transformation of
the image in **e** showing the absence of any additional superlattice spots. Both
**d** and **e** suggest that local chemical ordering is not present.

and B to NiCoCr-ODS appear to have improved the strength of the
alloy marginally. Notably, GRX-810 showed higher strength and duc-
tility when compared with the other ODS alloys; indeed, compared
with NiCoCr (where this study began[33]), GRX-810 provided twice the
strength and over three times the ductility, making it a much more
robust high-temperature alloy. One surprising result is the strength
of non-ODS GRX-810, which appears to be comparable to that of
as-built GRX-810 although it has limited ductility (comparable to the
non-ODS NiCoCr alloy). This finding suggests that the improvement in
strength is due to base composition, whereas the oxides are the source
of improved ductility. Additional alloys are compared in Fig. 3b, show-
ing the strength of GRX-810 and non-ODS GRX-810 compared with the
wrought Haynes 230 tested in this study (Supplementary Fig. 1), and

compared with wrought 625 and 718 from the literature[44]. Extended
Data Fig. 6a,b shows the as-built room temperature tensile tests. These
curves present little difference in regard to strength and elongation
between the different alloys, although GRX-810 did provide slightly
higher tensile strength compared with the three other alloys. The HIP
room temperature tests provide some variation in strength, because
the ODS alloys were able to retain higher strength after this process-
ing step. This is most probably attributed to the finer grain structure
maintained in the ODS alloys compared with the larger grain growth
and more equi-axed grain structure in the non-ODS NiCoCr sample[45].
Notably, the transverse ($x$–$y$) GRX-810 sample provided significantly
higher strength compared with those tested in the vertical ($z$) direction,
a typical result found with L-PBF materials[46]. This finding highlights the
anisotropy present in the AM samples that can not be recrystallized
through conventional means such as a HIP step, but also suggests that
print direction provides less strength than other orientations for these
ODS materials. Lastly, Extended Data Table 1 shows the tensile proper-
ties of as-built and HIP GRX-810 at varying temperatures. Two notable
observations are shown in this table. First, as-built GRX-810 consistently
offers higher strength compared with HIP GRX-810; and second, GRX-
810 provides unexpected cryogenic tensile properties (as-built GRX-810
provides 1.3 GPa of tensile strength), showing that nanoscale oxides are
not detrimental to alloy strength at these low temperatures. These high
cryogenic strengths have been noted in NiCoCr in past studies, and are
suggested to be due to a FCC-to-HCP phase transformation[13,20,23]. The
data in Extended Data Table 1 also show that GRX-810 remains ductile
from cryogenic to elevated (1,093 °C) temperature.

Creep tests were also performed at 1,093 °C to compare the prop-
erties of these alloys, and are shown in Fig. 3c,d. Figure 3c,d also
shows the impact of the combination of the oxide-strengthening and
model-driven composition of GRX-810 for high-temperature creep
strength. At 1,093 °C and 20 MPa, HIP GRX-810 ruptured after 6,500 h
of creep whereas the as-built test was terminated at 1% strain (over
2,800 h). All other non-ODS alloys considered, namely NiCoCr, AM
superalloy 718, AM superalloy 625 (at 14 MPa) and wrought Haynes 230,
ruptured in under 40 h. The orders of magnitude improvement in creep
performance by GRX-810 is also shown in Table 1, with time to reach 1%
strain at 1,093 °C under 20 MPa of stress for each alloy[5,47–49].

From Table 1 it can be seen that as-built GRX-810 required more than
500-fold longer to reach 1% strain compared with wrought Haynes 230,
and over 1,000-fold longer compared with AM superalloy 718. Also, as
observed from the tensile results, as-built GRX-810 exhibited better
high-temperature properties compared with HIP GRX-810. GRX-810
even provided better creep strength in this regime compared with
wrought Nb-based alloy C-103 tested in a high-vacuum environment[50].
At the higher 31 MPa stress level shown in Extended Data Fig. 6c,d,
as-built GRX-810 lasted for almost 2,500 h compared with NiCoCr,
which lasted just over 1 h–an almost 2,000-fold improvement in
life span.

One explanation for the improved tensile and creep properties of
GRX-810 may be the improvement observed in oxidation resistance
compared with Superalloy 718. In Fig. 4, the results from cyclic oxida-
tion tests conducted on GRX-810 and Superalloy 718 are shown up to
35 h at 1,100 and 1,200 °C. During exposure at 1,093 °C, the weight
loss observed for each alloy was attributed to oxide spallation on air
quenching from the test temperature. Nevertheless, the results shown
here indicate that GRX-810 has oxidative durability superior to AM
superalloy 718 at 1,093 °C, and significantly better at 1,200 °C, in which
AM superalloy 718 offered little to no life. More complete oxidation
analysis is provided in Extended Data Fig. 7.

## Comparison with current SOA AM alloys

Figure 3 and Extended Data Fig. 6 show that GRX-810 exhibits mark-
edly improved creep rupture properties over baseline NiCoCr

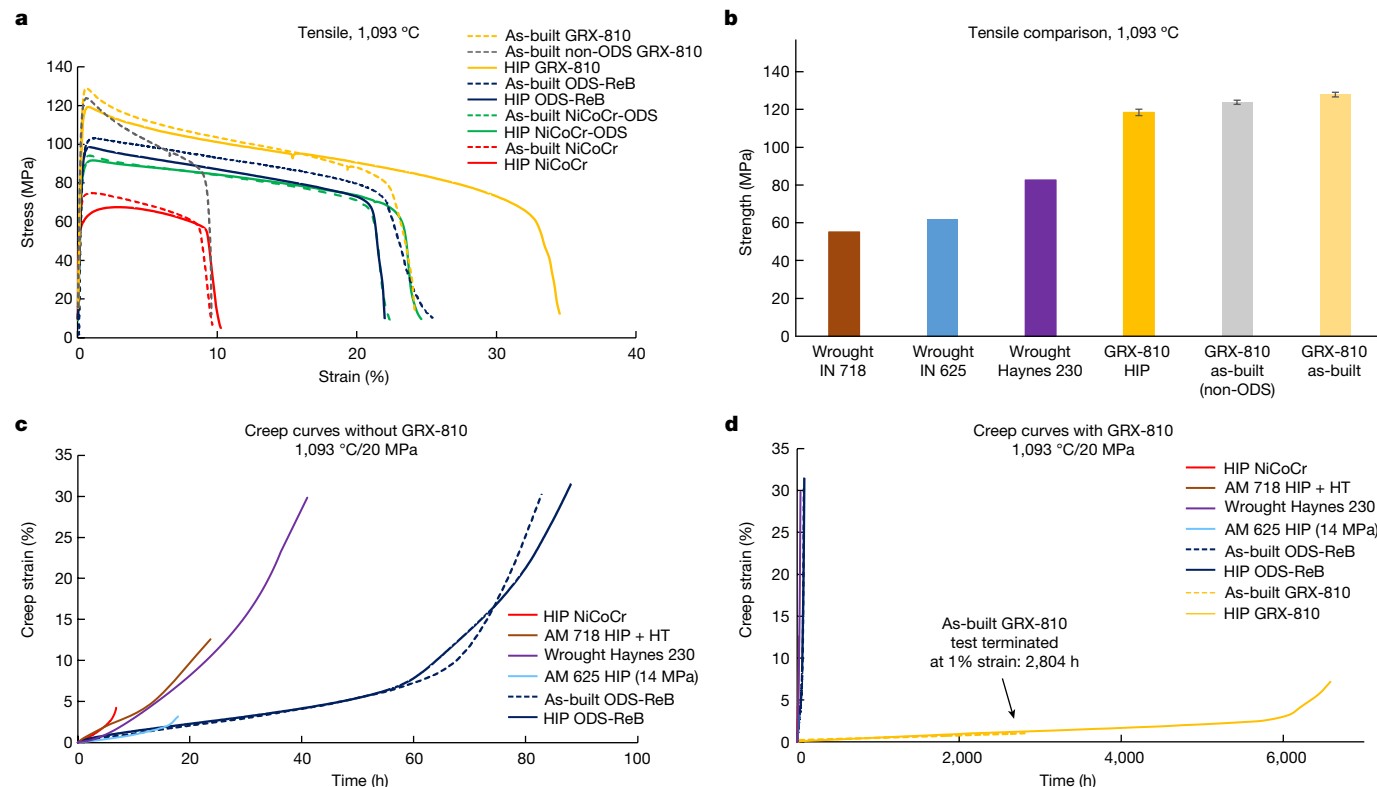

**Fig. 3 | Mechanical testing of NiCoCr-based alloys. a**, Engineering stress–strain curves at 1,093 °C for as-built and HIP alloys. **b**, Comparison of ultimate tensile strength between different alloys. Wrought 718 and 625 strengths were provided by the literature.[44] **c**, 1,093 °C creep curves for as-built and HIP NiCoCr, NiCoCr-ODS and ODS-ReB at 20 MPa. **d**, The same tests with GRX-810 curves included. Additional tests of AM 718, 625 and H230 are shown at 20 MPa for a better comparison with conventional high-temperature superalloys. Error bars correspond to 1 s.d.

**Table 1 | Time to reach 1% creep strain for GRX-810**

| Alloy | NiCoCr | AM 718 | AM 625[a] | Haynes 230 | ODS-ReB | C-103 (vacuum) | As-built GRX-810 | HIP GRX-810 |
|---|---|---|---|---|---|---|---|---|
| Time (h) | 0.35 | 2.2 | 10 | 5 | 9 | 1,170 (ref. 50) | 2,804 | 2,122 |

[a]Superalloy 625 testing was performed at 14 MPa[5,47–49].

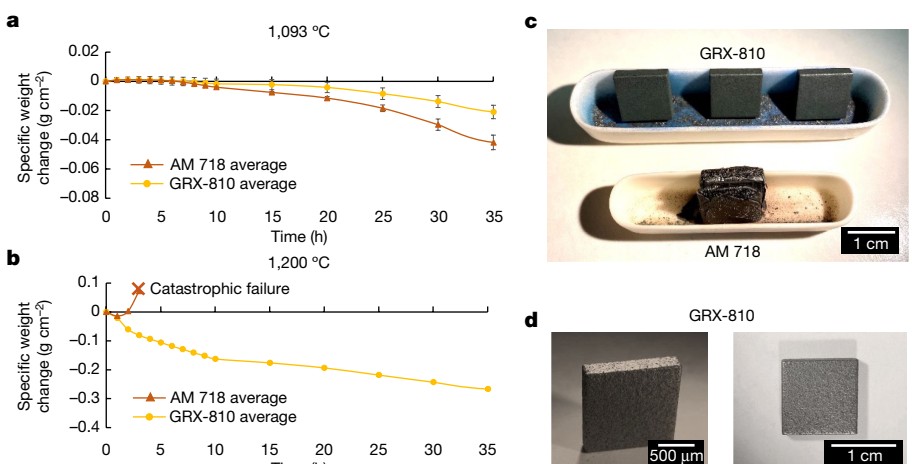

**Fig. 4 | Cyclic oxidation results at 1,093 and 1,200 °C. a,b**, Cyclic oxidation results for GRX-810 and superalloy 718 at 1,093 °C (**a**) and 1,200 °C (**b**) for up to 35 h. **c**, Optical images of oxidation samples after 100 h at 1,093 °C and 3 h at 1,200 °C, at which the superalloy 718 sample presented catastrophic oxidation. The three samples above are all GRX-810. **d**, GRX-810 samples after thermal cycling for 100 h at 1,093 and 1,200 °C. Error bars represent 1 s.d.

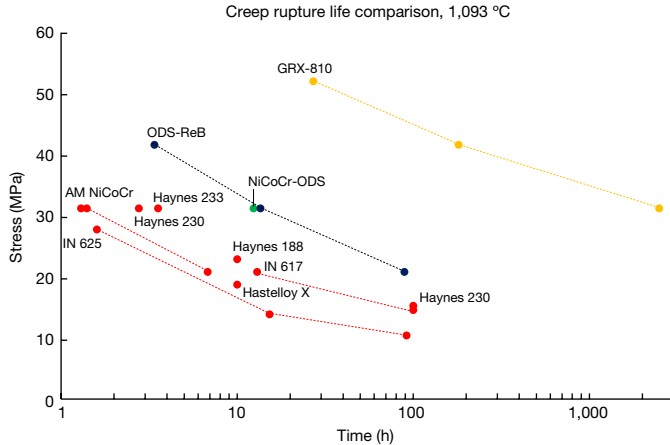

**Fig. 5 | Creep rupture life of as-built GRX-810 compared with current SOA AM superalloys.** Scatter plot of superalloy creep rupture life at 1,093 °C. GRX-810 presents superior creep properties compared with wrought alloys currently used in 3D printed high-temperature applications.

and NiCoCr-ODS alloys. Furthermore, compared with current state-of-the-art (SOA) AM high-temperature alloys (superalloy 718, superalloy 625 and Haynes 230), GRX-810 can provide orders of magnitude better creep life at 1,093 °C. To further illustrate this improvement, the 1,093 °C creep rupture lives of these alloys and other commercially available superalloys are plotted together in Fig. 5 (refs. 44,51–55).

The plot in Fig. 5 compares the high-temperature properties of both NiCoCr-ODS (green) and NiCoCr with Re and B additions (ODS-ReB) (blue), GRX-810 (gold) and conventional wrought superalloys used commonly in AM (red). In Fig. 3, although GRX-810 clearly shows improvement in tensile strength, its creep performance is even more pronounced and notable. Additional creep tests were performed, and results can be found in Extended Data Fig. 8. It is evident that the addition of nanoscale oxide dispersoids provided sufficient strength in the matrix to avoid dislocation motion (Supplementary Fig. 2), thereby leading to improvement in both mechanical and oxidation properties. However, STEM analysis of ODS-ReB and GRX-810 did not show any significant differences in oxide size or spatial distributions that could explain the differences in creep performance between the two alloys. Therefore, to better explain the creep strength of GRX-810, longitudinal sections taken from two 1,093 °C air creep tests at 20 MPa were analysed, as shown in Extended Data Fig. 9. Whereas other ODS alloys (for example, ODS-ReB) failed through a combination of grain boundary creep void coalescence and shear failure, GRX-810 appears to have suppressed these failure mechanisms because no apparent grain boundary voids/defects were observed after much longer test times. One contributory factor is that non-ODS GRX-810 has higher strength than even previous ODS alloys, and thus creep stress is a lower fraction of the alloy's yield stress. Nevertheless, the grain boundary failure modes in other ODS alloys suggest that in GRX-810 the stable MC carbides and solute segregation of W, Cr and Re along grain boundaries are factors contributing significantly to the protection of the alloy from grain boundary failure mechanisms. Previous studies have suggested that carbide stability at high temperature will influence grain boundary crack initiation during creep[56]. In addition, grain boundary diffusivity was reported to be correlated to the rate of void formation during creep[57,58]. Therefore, the addition of W and Re (known slow diffusors) should further inhibit creep void formation along grain boundaries whereas Cr segregation is expected to improve grain boundary corrosion and oxidation properties[59]. Stress-induced nitride formation was also observed in both the ODS-ReB (Cr-rich nitrides) and GRX-810 (Al- and Cr-rich nitrides) alloys. Whereas the formation of these internal nitrides is considered

detrimental to both alloys' properties[60], the nitrides in GRX-810 did not appear to contribute to grain boundary failure as observed in the ODS-ReB alloy.

In conclusion we present the design, characterization and properties of a new NiCoCr-based ODS alloy, GRX-810, which provides superior performance in extreme environments compared with current AM alloys. The use of computational modelling in alloy design led to a composition that balances properties and processability, with advanced characterization giving insights into the underlying microstructure and mechanisms. Creep performance of GRX-810 at 1,093 °C showed orders of magnitude improvement compared with currently used high-temperature alloys, thereby enabling the use of AM for complex components in extreme environments.

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

# Methods

## Materials

Three batches of pre-alloyed, gas-atomized powder feedstock (compositions given in Supplementary Table 1) were purchased from Praxair, Inc. The powders were sieved using +270 and −325 mesh (10–53 µm) to acquire an average particle diameter of around 15 µm as determined using a Horiba PSA300 Static Image Analysis System. The dispersoid used in the AM process was nanoscale $Y_2O_3$ powder (diameter 100–200 nm; American Elements). The powder was certified 99.999% pure yttrium oxide. These dispersoids were subsequently coated onto the base alloy powder using a high-energy acoustic mixer. Examples of powder morphology pre- and postmixed (coated), using the method described in ref. 33, are shown in Extended Data Fig. 1. Postmixed powder was then sieved using a 230 mesh screen to remove any large oxide or metallic powder particles. Unmixed powder (NiCoCr), mixed NiCoCr (NiCoCr-ODS) powder, NiCoCr-ReB (ODS-ReB), mixed GRX-810 and unmixed GRX-810 (non-ODS) samples were built using powder bed fusion to produce microstructural and mechanical test components on an EOS M100 L-PBF machine (40 µm beam diameter). For GRX-810 builds on the EOS M280, optimal densities were achieved with a laser energy density of 90–110 J mm$^{-3}$. Vertical test specimens (height 55.0 mm, diameter 6.35 mm) were built on 304 stainless steel build plates. All samples were then removed from the build plates using electrical discharge machining. Supplementary Table 1 provides a list of each alloy and its corresponding composition.

## Mechanical testing

After removal of test coupons from their respective build plates, selected specimens underwent a HIP cycle at 1,185 °C while wrapped in Ta foil to mitigate oxidation. The HIP cycle also had the benefit of removal of residual stresses. This provides a better comparison between ODS and non-ODS samples because residual stress has been shown to convolute structure–property relationships[61]. Both as-built and HIP specimens of each alloy type were tensile tested at room and elevated temperature using a cylindrical specimen with a 3.175-mm-diameter-gauge section. Cryogenic tensile testing at liquid N temperature (−196 °C) was also performed on GRX-810 samples. Testing was performed at Metcut Research, Inc. Tensile tests were performed at room temperature, at 0.127 mm min$^{-1}$ for the first 1.5% strain followed by an increase to 1.016 mm s$^{-1}$ until failure in accordance with the ASTM E8/E8M-21 standard, and at 1,093 °C at a constant strain rate of 1.016 mm min$^{-1}$ in accordance with the ASTM E21-17 standard. Following tensile tests, creep tests were performed at 1,093 °C by Metcut in accordance with the ASTM E139-11 standard. Testing of creep samples was continued until rupture (unless otherwise stated), after which they were then rapidly air cooled to maintain the fracture surface. All specimens were tested in the print direction unless otherwise specified in the description.

## Oxidation testing

Samples of AM 718 and the GRX-810 alloy were cut into samples of nominal size 12.5 × 12.5 × 3.5 mm$^3$, which provided a total surface area of around 487.5 mm$^2$. Surfaces were polished to a smooth finish with 1 µm diamond paste. Samples were oxidized in a laboratory airbox furnace at 1,093 °C for progressively longer dwell times. Exposure started at 1 h intervals for the first 10 h then at 5 h intervals for the next 25 h, followed by a 25 h duration and finally a 40 h duration for a total of 100 h at that temperature. Sample weights were measured after each interval for a total of 18 data points for each sample over the entire thermal exposure. After samples had reached the conclusion of the 100 h test at 1,093 °C, half underwent a second oxidation heat treatment at 1,200 °C for the same time period and intervals as the 1,093 °C test.

At 1,093 °C, the AM superalloy 718 and GRX-810 both underwent similar mass gain over the first few hours, indicating oxidation. However, both samples had exhibited mass loss by 7 h, which was accompanied by a spallation of oxide during each subsequent air quench to room temperature by removal from the box furnace. Specific weight change appeared to be linear in both AM superalloy 718 and GRX-810 from 5 to 10 h, with the rate of loss in the former roughly twice that of the latter. From 10–40 h, 5 h intervals were implemented and specific weight change per hour slowed, supporting the observation of oxide spallation during air quench to room temperature. The rate of loss in specific weight for AM superalloy 718 was again roughly twice that of GRX-810. Over the 25 and 40 h cycles shown in Extended Data Fig. 7, specific weight change rate slowed further and both alloys underwent equivalent weight change after air quenching to room temperature. A more significant level of spallation was observed at these longer intervals (as indicated by the larger drop in specific weight), but specific weight change per hour was lower than for the 1 and 5 h intervals.

After the 1,093 °C tests, half of the samples were removed and the remainder underwent additional progressive oxidation exposure at 1,200 °C following the same approach. In the case of AM superalloy 718, the sample lasted for only three 1 h cycles before undergoing catastrophic oxidation and complete disintegration and thus testing was ended. Runaway oxidation in the AM superalloy:718 sample can be seen in Fig. 4c, with significant weight gain observed after 1 h. The GRX-810 alloy exhibited similar behaviour to that at 1,093 °C exposure, although the specific weight change rate was around 40-fold more rapid over the 1 h heat treatment cycle. Over the 5, 25 and 40 h cycles, specific weight was only three- to fourfold more rapid than that at 1,093 °C over the same time intervals.

## Microstructural characterization

For SEM analysis, samples were polished using SiC grit paper followed by 0.5 µm diamond suspension. Afterwards, a final polish using 50 nm colloidal silica for 24 h was employed on samples used for electron backscatter diffraction (EBSD) analysis. EBSD orientation mapping was performed using an EDAX Hikari EBSD detector with 800 nm spot size. Postprocessing of maps was completed using TSL OIM Data Collection 7 software. High-resolution SEM imaging of the $Y_2O_3$ coating on NiCoCr powder was performed using a Tescan MAIA3 in ultrahigh-resolution configuration at 15 kV. Chemical maps were obtained with an Oxford Ultim Max Silicon Drift Detector and Aztec Software, and were used to determine phases in postcrept samples. STEM disc samples (diameter 3 mm) were extracted from metallographic samples of GRX-810 and ODS-ReB. STEM samples were thinned manually to 130 µm using 600 grit SiC polishing paper. To achieve electron transparency, polished STEM discs were electropolished with a solution of 90% methanol and 10% perchloric acid at −40 °C and 12 V using a Struers twin-jet polisher. Microstructural analysis was performed on an FEI Talos at 200 kV using a HAADF detector. Defect analysis was performed using S-CORR probe aberration-corrected and monochromated Thermo Fisher Scientific Themis-Z STEM at an acceleration voltage of 300 kV. STEM diffraction contrast imaging was performed with BF and HAADF detectors by selection of the appropriate camera length. Atomic resolution of the microstructure was performed by tilting of the thin disc foils into specific low-index crystallographic zones. High-resolution EDS data were collected by a Super-X energy dispersive X-ray spectroscopy detector in Themis-Z. Data were collected and processed using Thermo Fisher Scientific Velox software. In particular, raw data in the original spectral maps were quantified using standard Cliff–Lorimer (k-factor) fit (default k-factors available in Velox were used, as well as the Brown–Powell empirical ionization cross-section model) including background subtraction. STEM micrographs were corrected for potential sample drift and scanning beam distortions using the drift-corrected frame integration function of Velox.

The Archimedes method with deionized water as the immersion fluid was used to determine the density of the additively manufactured GRX-810 material. Specimens with surface-breaching networks of cracks

and porosity allow water infiltration that can be observed as bubbling during submersion; no bubbling was observed during submersion of GRX-810. Measurements of a part's mass in air ($M_a$) and in water ($M_w$) were taken on a Mettler Toledo XS205 system. Density of the AM part was calculated by

$$p = \left(\frac{M_a}{(M_a - M_w)}\right) \times (p_w - p_0) + p_0 \qquad (1)$$

where $p_w$ is the temperature-dependent density of water and $p_0$ is air density. The reported density value is an average of three independent measurements.

Powder feedstock (15–45 μm) of the optimized composition shown in Fig. 1 was obtained, coated with $Y_2O_3$ nanoparticles and built with L-PBF using the steps detailed above. Successful production of GRX-810 using AM L-PBF allowed for the characterization of GRX-810 in both as-built and post-HIP states. Extended Data Fig. 2 shows the high density (over 99.97%) that can be achieved with optimized print parameters for GRX-810 based on optical microscopy analysis. Relative density measurements further confirmed this value, showing 99.96% density value for the same sample. From the SEM analysis shown in Extended Data Fig. 5, the major difference in microstructure between as-built and HIP GRX-810 samples is the presence of fine MC carbides along grain boundaries after the HIP processing step. These grain boundary phases were confirmed as Ti/Nb-rich carbides through both SEM and TEM EDS, for which the results of the former can be found in Supplementary Fig. 3. No other phases were present in either material state, validating the accuracy of thermodynamic calculations in prediction of stable microstructures in the compositional space. The intragranular dark contrast 'dot-like' features observed in Extended Data Fig. 5b are $Y_2O_3$ particles sufficiently large to be observed by SEM using secondary electron imaging mode. The lack of bulk oxide formation provides further evidence that the coated GRX-810 powder can be successfully printed using L-PBF to form an optimized oxide dispersion-strengthened alloy.

From Extended Data Fig. 4a, little to no variations in grain structure and average grain diameter between as-built and HIP conditions were observed in GRX-810. This finding suggests that the distribution of fine oxides sufficiently suppresses dislocation and grain boundary movement at high temperatures. The grain texture usually associated with AM melting processes is apparent between the $x$–$y$ and $x$–$z$ planes[23,46]. Extended Data Fig. 4b illustrates LAADF–STEM DCI showing defect configurations and corresponding EDS chemical maps. The yttrium map confirms the presence of $Y_2O_3$ particles evenly distributed throughout the GRX-810 matrix. In fact, this distribution of oxides appears to have pinned the dislocations produced during the L-PBF build step as a high-dislocation density that can be observed in the LAADF–STEM DCI micrograph. The Cr and Ni maps represent chemical maps of the other elements, showing no local segregation or ordering at 500 nm length scale.

## Modelling of the next-generation MPEA

Thermodynamic modelling (CALPHAD) was employed to produce a superior composition using equi-atomic NiCoCr as a foundation[62,63]. Simulations were completed adding specific elements (for example, B, C, Al, Ti, V, Mn, Fe, Zr, Nb, Mo, Hf, Ta, W and Re) to an equi-atomic composition of NiCoCr. Therefore, as new elements were included in the simulation, the atomic percentages of Ni, Co and Cr remained equal. Thresholds were used to better constrain the models and guide optimization. The thresholds included (1) maximized solid solution strengthening; (2) the FCC solid-solution matrix should be maintained; for this constraint it was assumed that any undesired phases stable above 810 °C would not be acceptable; (3) enabling of MC carbide formation along grain boundaries stable above 1,200 °C; and (4) the temperature difference associated with a composition solidification temperature range (STR) must remain below 100 °C for AM printability.

This reduction in STR has been used by welding engineers to predict an alloy's susceptibility to solidification cracking, dendritic segregation (which requires remedial postprocessing), residual stress and hot cracking[33,64]. These constraints allowed the alloy compositional space to be manageable and reduced the number of overall simulations, thereby leading to the optimized composition and predicted equilibrium phases shown in Fig. 1a. Simulations were performed using Thermo-Calc v.2020b with the Ni alloy database TCNI8. Upwards of $10^7$ equilibrium calculations were performed across composition and temperature space. Y and O were not considered in the composition search, because the $Y_2O_3$ phase is expected to be inert and is not well described in the TCNI8 database[62]. The $Y_2O_3$ line shown in Fig. 1 is approximate and is included here for visual clarity.

To better understand the phase stability and properties of the NiCoCr compositional space, a full overview using DFT calculations was performed[65]. Extended Data Fig. 3 shows the computed electronic spin density of states of the FCC (A1), BCC (A2) and HCP (A3) phases of the equi-atomic NiCoCr alloy system. Interestingly, from Extended Data Fig. 3, the formation energy of equi-atomic NiCoCr (relative to elemental solids) is positive. At this composition, energies of the FCC and HCP phases are quite comparable, suggesting there is a crossover from the HCP to FCC ground state nearby. Both FCC and HCP phases are close packed and the computed equilibrium volumes per atom for FCC and HCP at any composition are rather close, with the equilibrium volume of BCC differing significantly. In the close-packed phase (such as FCC), the stacking-fault energy increases with the absolute value of the energy difference $E$(HCP) − $E$(FCC). Creep properties are influenced by stacking-fault energy, which depends on composition. For future guidance on NiCoCr-based alloy development, the results from Extended Data Fig. 3 are organized into a predicted ternary phase diagram at 0 K in Fig. 1b. Although this phase diagram may not represent stable phases at high temperatures due to entropy, these calculations have important implications in the properties of the NiCoCr system at cryogenic temperatures. Recent papers have found excellent mechanical properties for NiCoCr-based medium-entropy alloys at these low temperatures in which the phase transformation from FCC to HCP during deformation is the principal contributing factor[23,66]. Therefore, these low-temperature properties may be further improved by moving the composition of NiCoCr into the more stable HCP phase regime while maintaining an FCC phase. Future work is planned to explore this possibility.

## DFT calculations

For this study, the all-electron DFT KKR-coherent potential approximation (CPA) Green's function code[67] was employed to calculate energies of disordered structures. Within the KKR method[68,69], atomic sphere approximation[70] was used with periodic boundary corrections[71]. The basis of atomic orbitals within the atomic sphere approximation spheres included $s$, $p$, $d$ and $f$ orbitals ($l_{max}$ = 3). In addition, the PBEsol GGA-type exchange correlation functional[72] was used. Self-consistency was achieved using a modified Broyden's second method[73]. Integration in a complex energy plane was performed using the Chebyshev quadrature semicircular contour with 20 points. Homogeneous atomic disorder was addressed by CPA[65]. Crystal structures considered included A1 (FCC), A2 (BCC) and A3 (HCP). The ideal c/a = $(8/3)^{1/2}$ = 1.632993 was chosen for the HCP phase. A special k-point mesh[74] for Brillouin zone integration included $18^3$ k-points for FCC and BCC one-atom primitive cells and 16 × 16 × 8 for the HCP two-atom unit cell; an auxiliary secondary $12^3$ mesh was used for FCC and BCC and 10 × 10 × 6 for HCP in KKR-CPA code[67].

## Data availability

The experimental data that support the findings of this study are available from the corresponding author on reasonable request. Source data are provided with this paper.

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

**Acknowledgements** Funding for this study was provided by NASA's Aeronautics Research Mission Directorate—Transformational Tools and Technologies Project Office and NASA's Space Technology Mission Directorate Game Changing Development Program under the optimized and Repeatable Components in Additive Manufacturing project. M.H. and M.J.M. acknowledge the support of the National Science Foundation and the DMREF programme under grant no. 1922239. For more information on this technology, and to discuss licencing and partnering opportunities, please contact grc-techtransfer@mail.nasa.gov and reference LEW-19886-1 and LEW-20020-1.

**Author contributions** T.M.S. wrote the manuscript. T.M.S., C.A.K., T.P.G. and P.R.G. designed experiments and performed microstructural/mechanical characterization. T.M.S., M.H. and M.J.M. performed TEM analysis. T.M.S. produced the powder feedstock, both coated and uncoated. A.C.T. operated the EOS M100 and developed the build parameters. B.J.H. performed cyclic oxidation tests. C.A.K., N.A.Z. and J.W.L. performed the CALPAHD and DFT models.

**Competing interests** The authors declare no competing interests.

**Additional information**
**Correspondence and requests for materials** should be addressed to Timothy M. Smith.

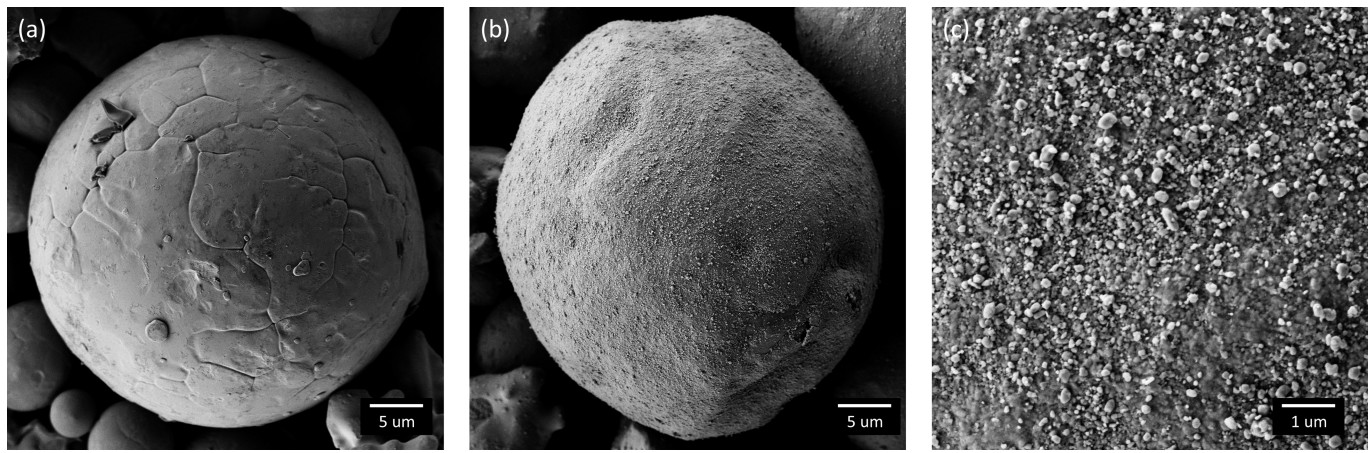

**Extended Data Fig. 1 | Powder Characterization. a**,**b**, Secondary electron SEM images of an uncoated ReB metal powder particle (**a**) and a coated ReB metal powder particle (**b**). **c**, A higher-resolution image of the coating in **b**.

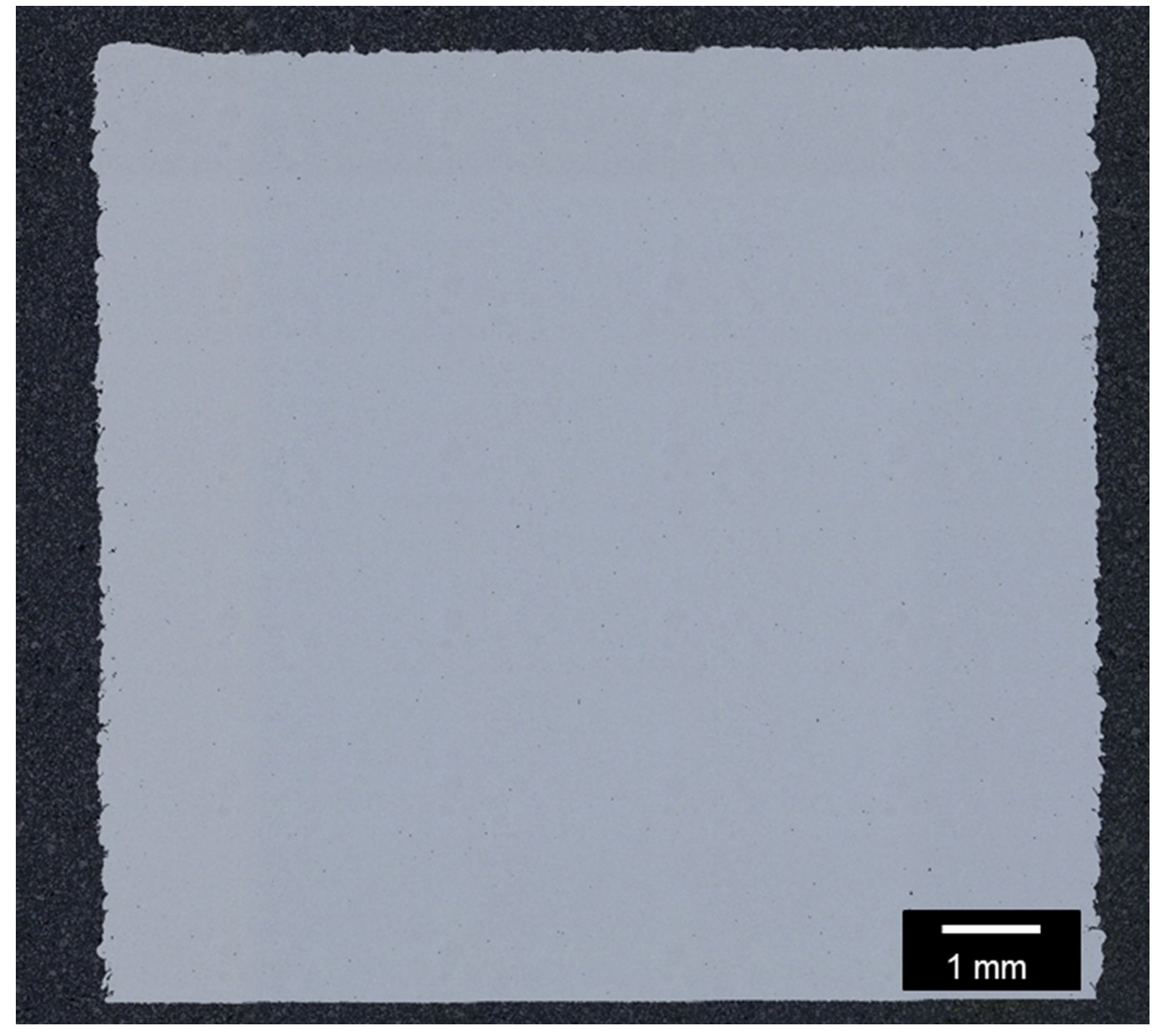

**Extended Data Fig. 2 | Defect characterization of As-built GRX-810.** Optical microscopy image of as-built GRX-810 with optimized print parameters. Image segmentation analysis suggests that the density of the as-built part is above 99.97%.

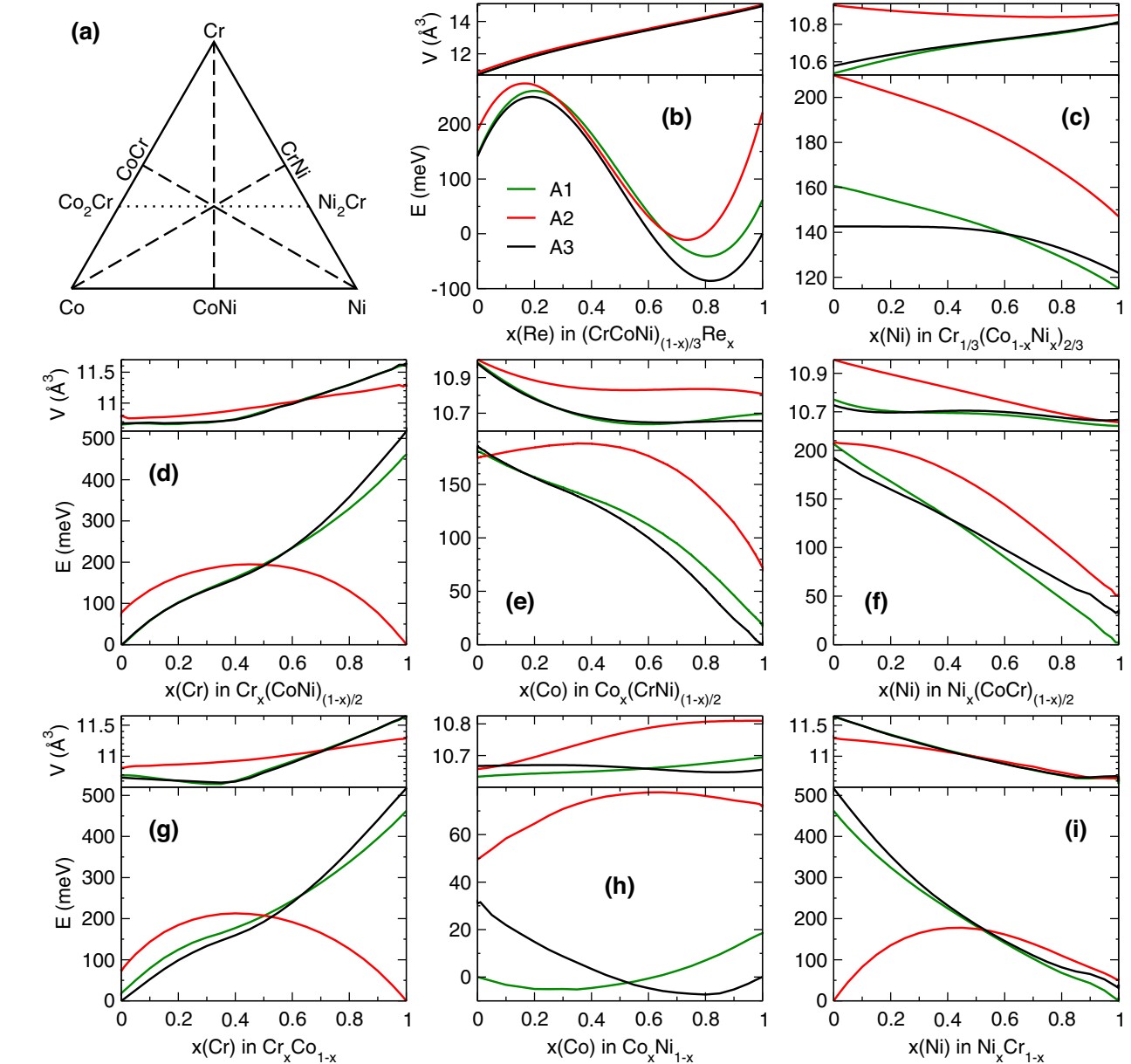

**Extended Data Fig. 3 | Density functional theory computed formation energies of the Ni-Co-Cr system. a–i**, Computed formation energies of A1 (FCC), A2 (BCC), and A3 (HCP) phases at zero pressure versus composition along the selected lines (**a**) for: **b**, quaternary $(NiCoCr)_{(1-x)/3}Re_x$; **c–f**, ternary; and **g,i**, binary alloys.

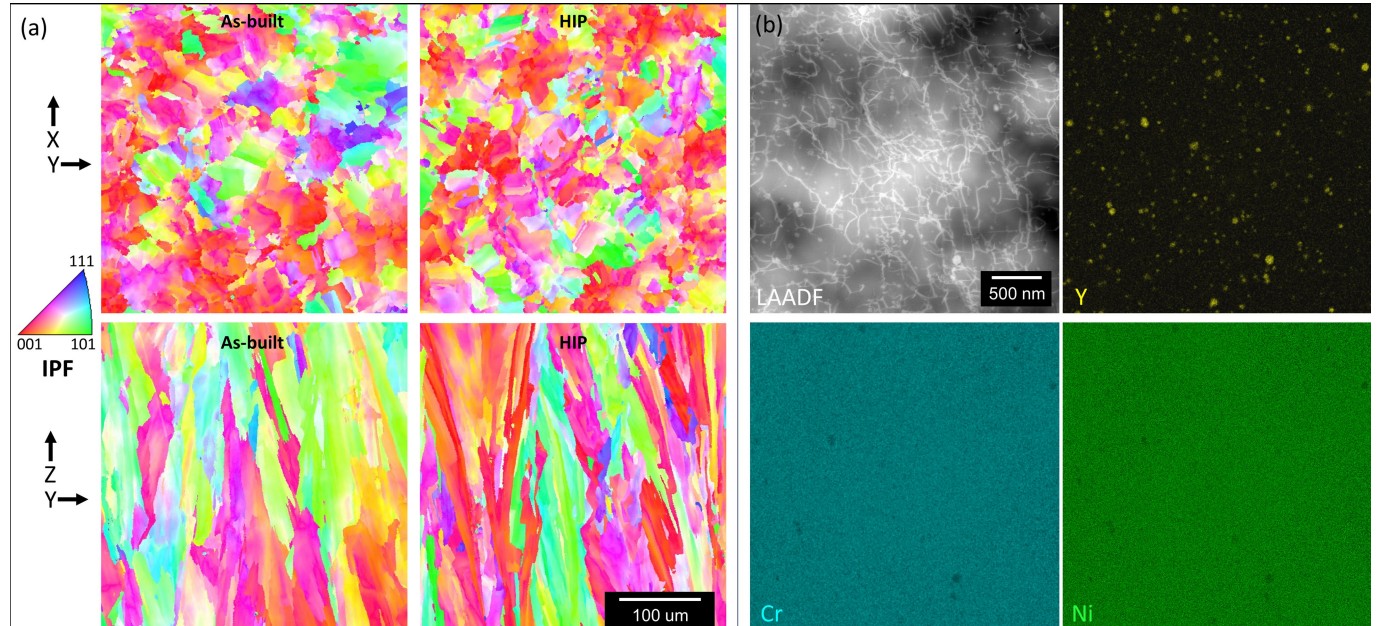

**Extended Data Fig. 4 | Microstructural characterization of HIP GRX-810.**
**a**, Electron back scatter diffraction (EBSD) inverse pole figure maps of the X-Y build plane and the Y-Z build plane where the Z-axis represents the build direction. Represented are maps from as-built and HIP samples. No significant difference was observed between the as-built and post-HIP grain structure. **b**, STEM-EDS Y, Cr, and Ni maps and corresponding LAADF-DCI micrograph of untested HIP GRX-810.

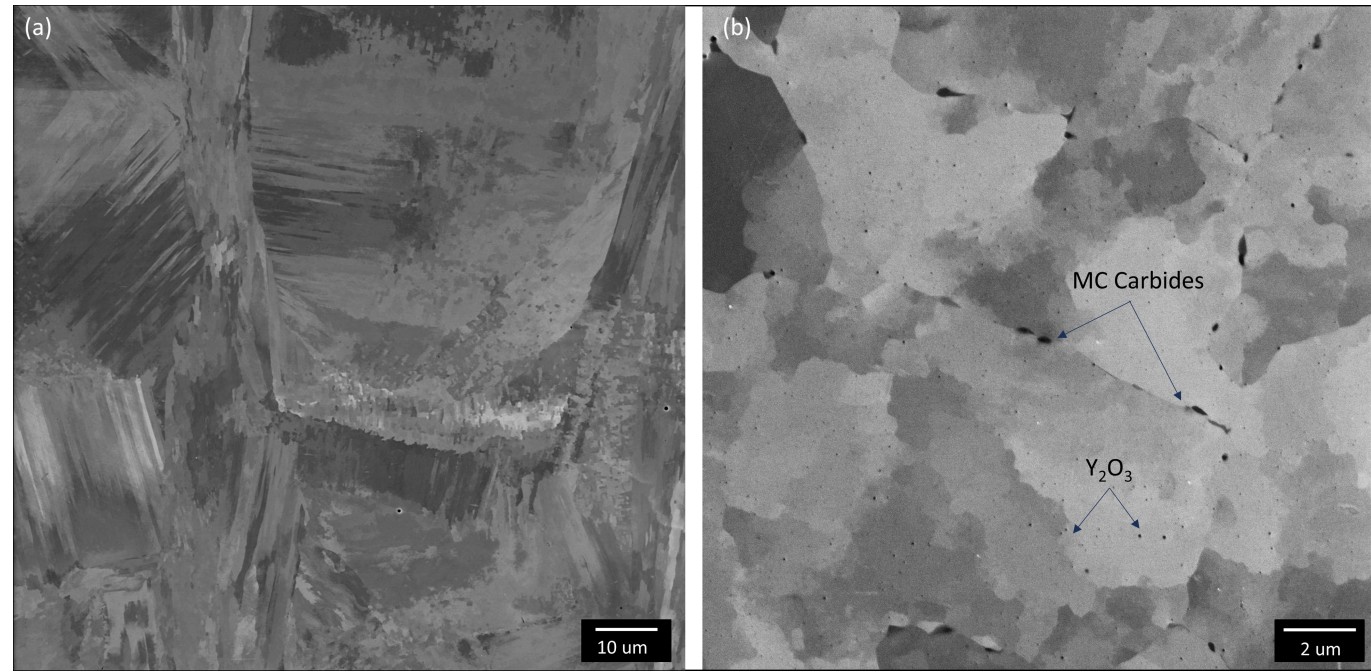

**Extended Data Fig. 5 | Scanning electron microscopy analysis of as-built and HIP GRX-810. a,b,** Secondary electron SEM images revealing the microstructure of (**a**) as-built GRX-810 and (**b**) HIP GRX-810 transverse to the build direction (X-Y). In the higher resolution image in **b**, both MC carbides and nanoscale $Y_2O_3$ particles can be observed.

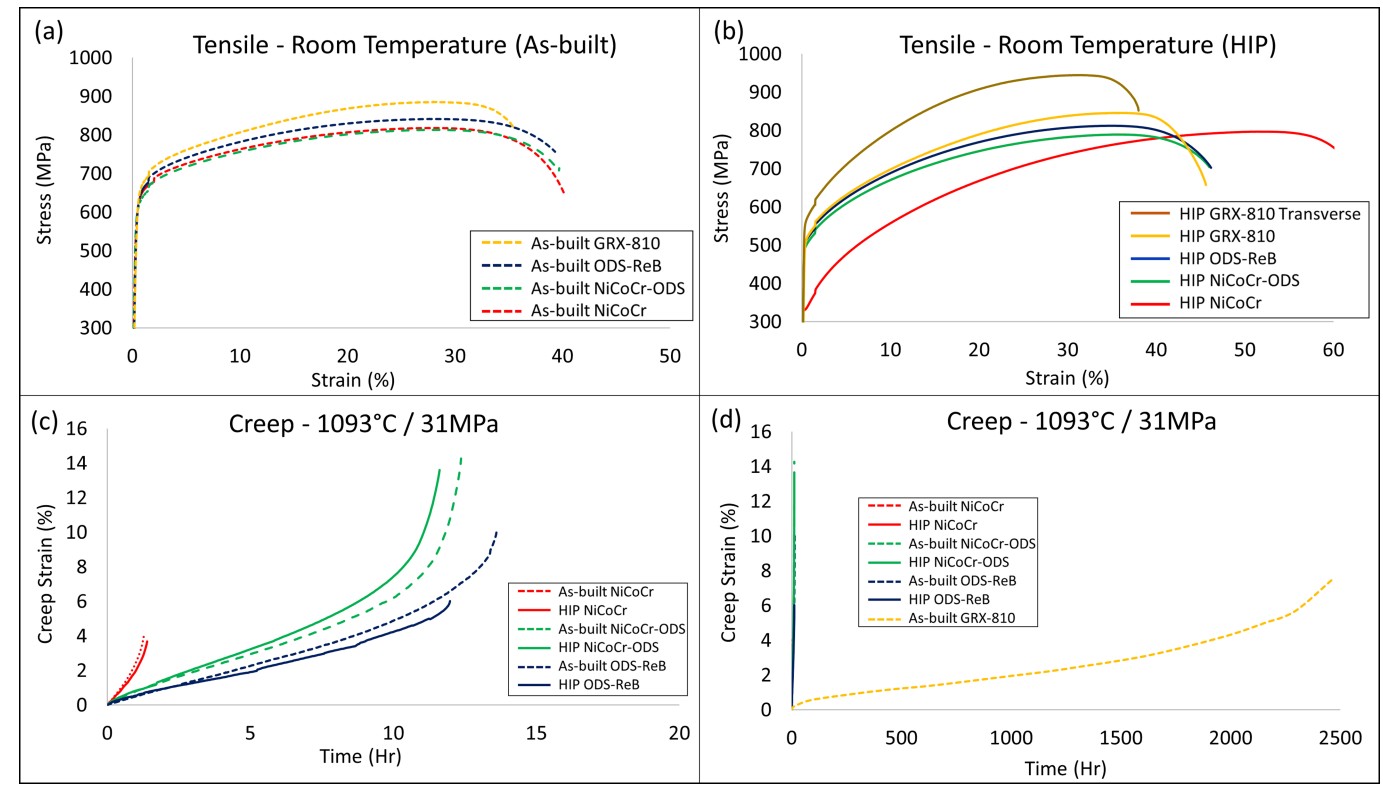

**Extended Data Fig. 6 | Mechanical tests of NiCoCr-based alloys.**
**a**, **b**, Engineering stress-strain curves at room temperature for the as-built alloys (**a**) and for HIP alloys (**b**). The step between 1% and 2% strain results from an increase in the tensile strain rate which is consistent with ASTM E8 standard. **c**, 1,093 °C creep curves for as-built and HIP NiCoCr, NiCoCr-ODS, ODS-ReB at 30MPa. **d**, The same tests with GRX-810 curves included.

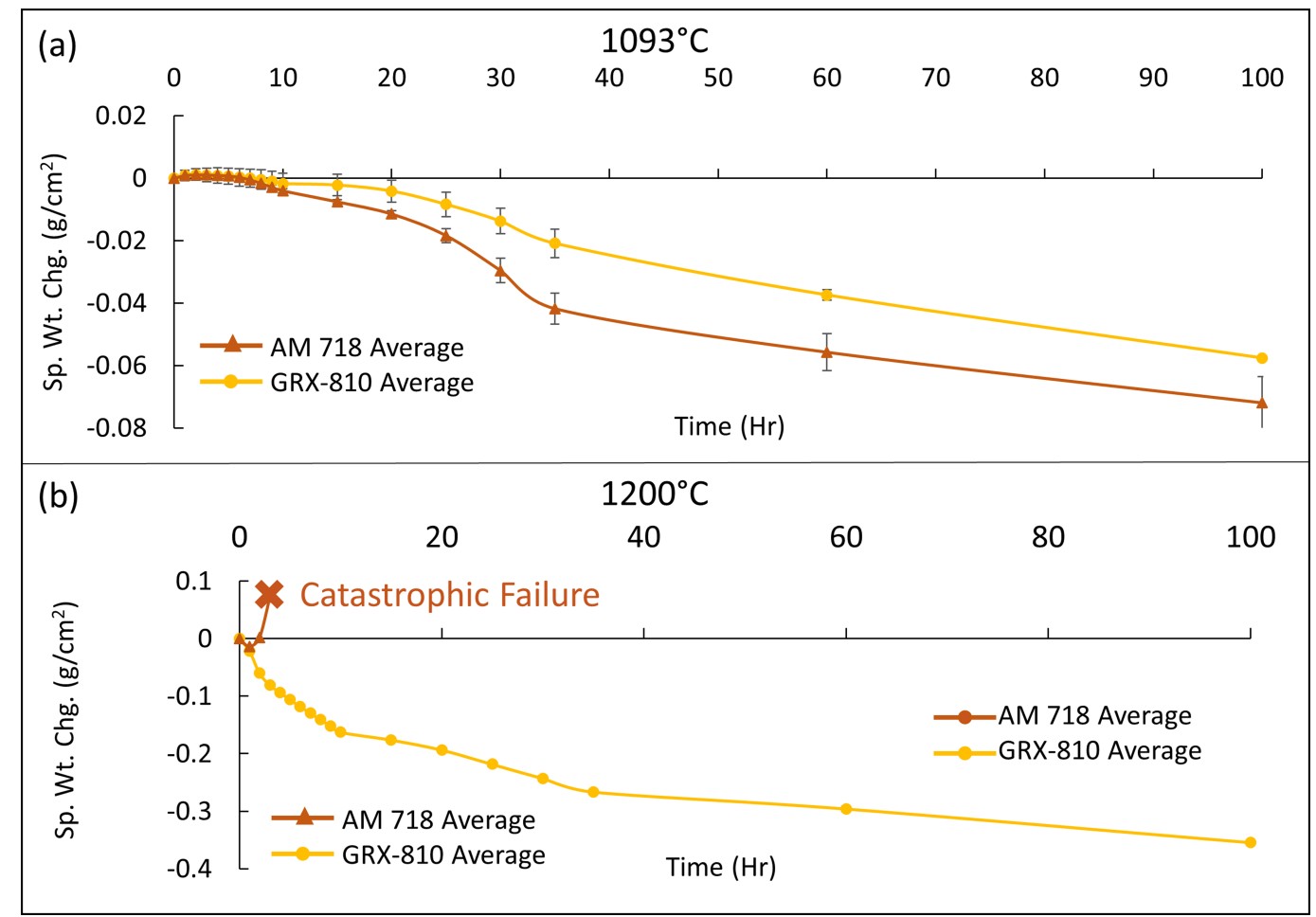

**Extended Data Fig. 7 | Cyclic oxidation results at 1,093 °C and 1,200 °C. a,b,** Cyclic oxidation results for GRX-810 and superalloy 718 at 1,093 °C (**a**) and 1,200 °C (**b**) up to 100 h. Error bars correspond to 1 standard deviation.

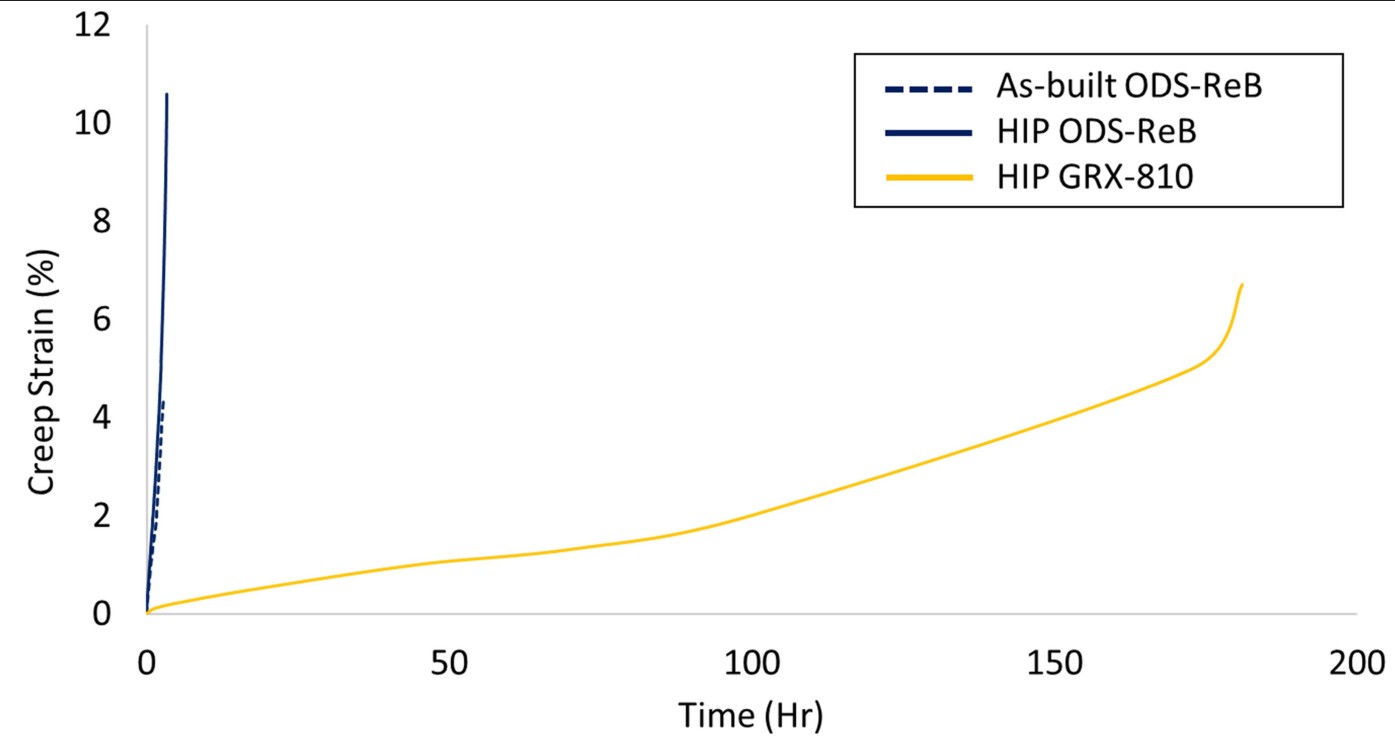

**Extended Data Fig. 8 | Creep results at 1,093 °C / 41MPa.** Creep results at higher stress. Creep curves of ReB-ODS and HIP GRX-810 at 41 MPa / 1,093 °C.

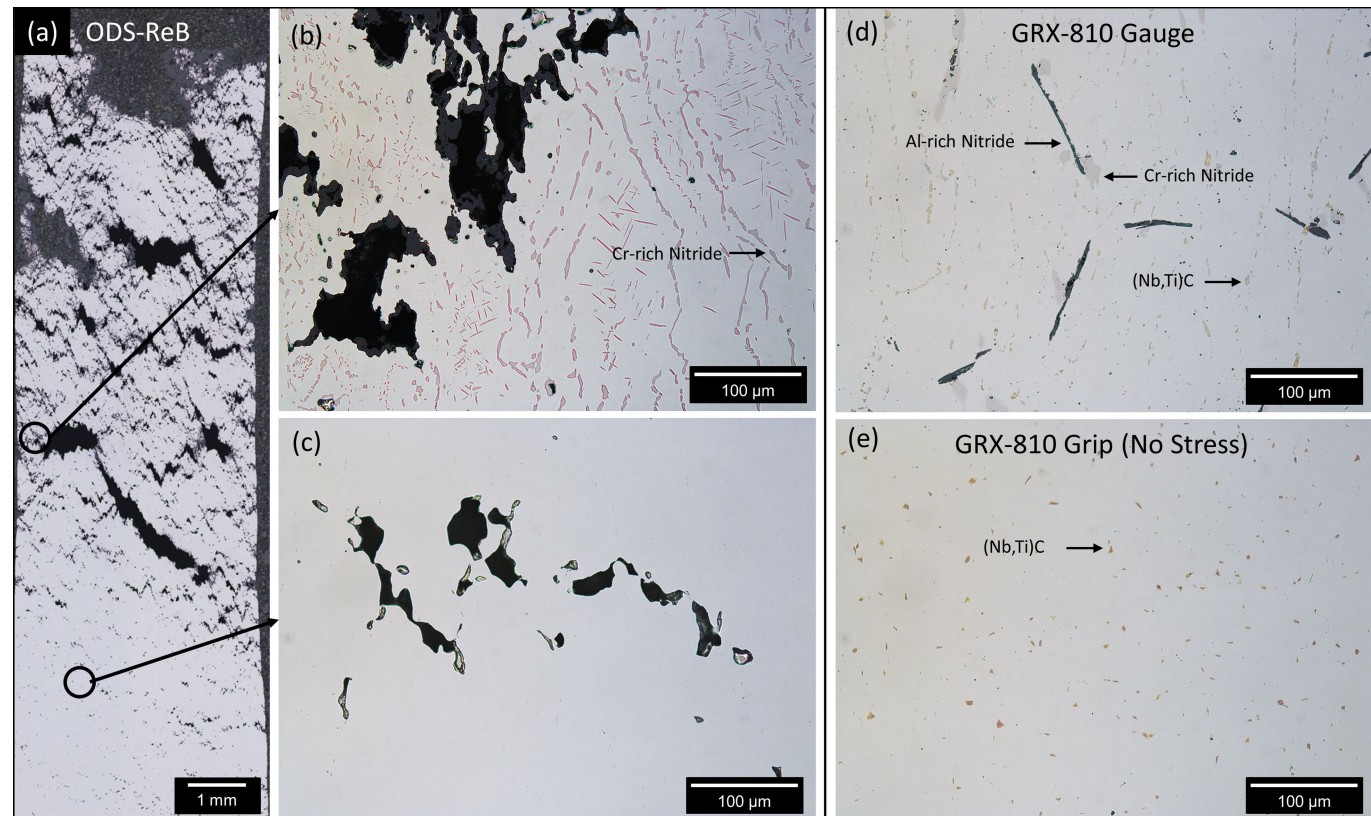

**Extended Data Fig. 9 | Optical analysis of creep deformation. a**, An optical cross-section of the as-built ODS-ReB sample tested at 1,093 °C / 20 MPa. **b**, Creep pores/overload cracking along with a Cr-rich nitride in a high plasticity region of the sample. **c**, A region removed from the fracture surface that reveals creep void formation and lack of nitride formation. **d**, A representative micrograph of the microstructure in the as-built GRX-810 sample tested at 1,093 °C / 20 MPa that was terminated at 1% strain after 2,800 h. No creep void formation is observed but the presence of Al-rich and Cr-rich nitride phases along grain boundaries can be seen. **e**, A representative micrograph of the microstructure from the grip section of the same as-built GRX-810 sample revealing that nitrides did not form in an area under no stress. These results suggest that the nitride phases are creep induced.

**Extended Data Table 1 | Overview of tensile results for As-built and HIP GRX-810**

| Temperature (C) | As-Built Tensile Strength (MPa) | As-built Yield Strength (MPa) | As-built Elongation (%) | HIP Tensile Strength (MPa) | HIP Yield Strength (MPa) | HIP Elongation (%) |
|---|---|---|---|---|---|---|
| -195.6 | 1,303.1 | 910.1 | 39.6 | 1,227.3 | 723.9 | 49 |
| 21.1 | 882.5 | 641.2 | 33 | 848.1 | 515 | 43 |
| 426.7 | 710.2 | 527.4 | 33.3 | 655 | 410.2 | 40 |
| 648.9 | 675.7 | 479.2 | 32.1 | 630.9 | 368.9 | 43 |
| 871.1 | 292.3 | 249.6 | 56.1 | 262.7 | 206.2 | 62 |
| 1000 | X | X | X | 164.1 | 161.3 | 44 |
| 1093.3 | 128.9 | 127.6 | 22 | 119.3 | 115.8 | 32 |

Tensile results of as-built and HIP GRX-810 at varying temperatures.