## [Peer Review File · Nature]

Manuscript Title: A 3D Printable Alloy Designed for Extreme Environments

Reviewer Comments & Author Rebuttals

Reviewer Reports on the Initial Version:

Referees' comments:

Referee #1 (Remarks to the Author):

Some general comments:

Principally a good idea to develop an oxide dispersion strengthened (ODS) NiCoCr-based alloy (although it should be shown a bit more clearly, what the specific progress in the state of the art is compared to previous papers, pls see comments below).

Otherwise the work seems to show a really nice and big leap forward: 2X improvement in strength, over 1000X better creep performance, and 2X improvement in oxidation compared to traditional Ni based alloys at 1093 °C are impressive features indeed – maybe state here to WHICH Ni alloy / alloy group this comparison refers to (because the variety in properties of such alloys is quite large; which material is the closest competitor etc) ? (this is more important than coining a new name here in the abstract, that can be done later in the text). Also, pls give details of the design process (see several comments below), because 'ICME did the job' does not tell so much.

In general I find that the paper should be considered for possible publication. The results and the progress are impressive.

Further items / details:

1. Using / exploiting the in-situ reaction of oxygen with specific alloying elements for material design in additive manufacturing was shown a few times before in literature and hence it should be made clear what the specific novelty here is compared to the state of the art before.
2. The sentence 'These results represent a paradigm shift in alloy development, where ICME combined with AM can be leveraged to accelerate the development of revolutionary materials' is maybe a bit too bombastic (not really necessary here?) – if the results are so strong as they are, better be more specific on the exact scientific / engin. novelty items; better work out / state what is the more specific innovation – how can I translate to other creep alloys too etc – maybe be a bit more specific here. What is the key innovation – is it the fine dispersion ? Is it to do that in-situ during AM; is it to control the O₂ partial pressure during AM etc ? Better be more specific here on the innovative and inventory height – this is more helpful than a less specific 'sales pitch'.
3. The inventory advancement of the topic / field could be highlighted a bit more convincingly: A few papers were published before on the use of Y-based oxide dispersions in such materials, also with high radiation resistance etc.; Example: High radiation tolerance of an ultrastrong nanostructured NiCoCr alloy with stable dispersed nanooxides and fine grain structure, by C Lu et al. In the Journal of Nuclear Materials showed in 2021 very good behaviour of nanostructured NiCoCr medium entropy alloys enhanced by Y-Hf-O nanooxides etc – so, similar as the comment above: better describe a bit more clearly what the actual item(s) of highest progress is, maybe the combination of AM and creep properties etc ?
4. Readers might not know what ICME is. Recall this is for general readers. Is the term here really required or rather distracting?
5. In the introduction, the authors report in general terms on the particularly favourable deformation mechanisms in the family of Cantor alloys and related materials (strength / ductility trade off etc)- but it seems that these properties are not quite so relevant to the alloys discussed

here, since these features are room-temperature properties? But this paper is about creep ? Maybe check storyline in the intro ?

6. Not good to use s.t. like 'GRX-810' in a paper for a general science readership; readers do not know what that is / means. Better pick more general title.

7. The naming convention 'This new ODS alloy 20 called GRX-810 (Glenn Research Center Extreme Temperature above 810 °C)' is maybe a bit too specific for an abstract of such a journal ?

8. Details missing for refs 39, 46, 47, 50 etc ? better check all refs.

9. Ref 54: the name's not 'Schrodinger'...

10. Ref 12 was not the first paper to show that the strength-ductility trade-off can be overcome as a result of atomic-scale deformation mechanisms. This was reported in earlier papers before for these alloys.

11. The use and beneficial effects of C and other interstitials on HEAs was shown in several papers before the here quoted ref 22 – maybe best double check to use the original references also ? (I know, not easy in this fast moving field).

12. The authors use the term 'ICME approach' and it seems here that they essentially mean by that that they did phase diagram calculations prior to synthesis, hence, maybe just better state exactly that ? The ICME terminology for doing s.t. that is well established is maybe a bit distracting? I would tend to be much more exact and specific in the method / design description and avoid any buzz terminology that is probably unsuited for general readers to follow? The results section is much more clear here, so maybe consider rewriting the intro section scientifically a bit more specific and maybe state 'thermodynamics calculations or so' instead of 'ICME' ?

13. For example they write '...an ICME approach was employed to design a NiCoCr system for high temperature applications using AM for complex components. This effort resulted in a new composition ...' What exactly was done, which steps, which simulation etc. ? The term ICME does not really say anything about the science that was done ? I guess it was a phase diagram and a Scheil simulation or so using ThermoCalc and Dictra or so ? Better be more specific here in such paragraphs?

14. Statement 'slight additions of Re and B'...? better be more specific – these dopants are quite essential – which range was explored – these are essential information items for understanding the design process. Also, add a Fig reference here. Would be nice to have a bit more specific details in some of these statements.

15. Statement 'This study confirms the maturity of both ICME driven alloy design...' Not sure – it was neither explained what ICME is nor was stated what exactly was done ... ICME is more a very general buzzword but the authors need to specifically state how the design was done, which TD / kin. calculations were done etc using WHICH database etc, screening which range of interstitials etc – all this info must be briefly added here, as just stating 'ICME' did it all is not informative.

16. Term 'alloy "trade space' is unclear.

17. Paragraph on DFT and ff: 'a full overview using density functional theory (DFT) calculations was performed...': be more specific here: were these ground state simulations of free energy simulations (for phases, SFs etc) ? (a) If the latter is true, which entropy models were used ? (b) if the former is true, the 0K results must be more critically discussed w.r.t. relevance. Fig. 1 seems to suggest we look at 0K data from DFT ? This point in the overall design approach is not quite so clear as the authors develop a HIGH-temperature alloy, where entropy is surely essential ? Hence, what is the point of discussing 0K results here and what is the expected error / effect for the here targeted high temperature case? Needs a bit deeper discussion I suppose.

18. Powder production: was the material pre-alloyed and WHICH composition was pre-alloyed ? I assume the pre-alloyed powder were low in O and the O came into play during AM ? Maybe give more info here on the intermediate states ?

19. The stacking fault discussion / info regarding Fig 3 is from a room temperature image – what is the estimated role / existence of the SFs and SF tetrahedra at operation temperature / high temperatures ? Do these features matter for creep or are they mere ambient temp features ?

20. Fig. 4: better to add reference data from a few established (commercial) Ni base superalloys into the same diagram ? Non-expert readers might find it otherwise hard to see the differences to established alloys in this field ? (similar as nicely summarized in Fig 7 – this seems to be the real key fig of the paper ? maybe add some / a few of these ref data also to Fig 4 ?)

21. Some abbreviations and material names used in the legends etc are unknown to general readers, such as e.g. H230 etc – all should be clearly given / explained etc.
22. In Fig 7 (and all figs): pls make sure that ALL abbreviations / alloy names etc are explained somewhere in the paper / overview table etc / give compositions etc – that would be really helpful.

Referee #2 (Remarks to the Author):

In this manuscript and building upon their previous research, Smith et al. present a high-quality report on the microstructure and high temperature performance of the oxide dispersion strengthened, NiCoCr-based, multi-principal element alloy GRX-810. The alloy was designed using an integrated computational materials engineering approach, which facilitates rapid alloy development. The compositional design requirements were such that a i) face-centred solid solution should be maintained, ii) the freezing range of the alloy should be 100°C or less to improve processability via additive manufacturing, iii) metal carbides can form along grain boundaries, which are stable above 1200°C and that iv) solid-solution strengthening should be maximised. The ICME approach used in this study falls outside my area of expertise, so I will discuss the following outstanding features of this original piece of work in the attached document. Please refer to the attached reviewer's report (Word document and PDF version) as this contains equations and figures, which I am unable to include in text format.

Referee #3 (Remarks to the Author):

The work focuses on a Y2O3 oxide dispersion strengthened (ODS) NiCoCr alloy termed GRX-810 that has been processed using laser powder bed fusion (LPBF). The material appears to have been designed using a combination of CALPHAD and DFT aiming to maximize solid solution strengthening while retaining a primarily FCC structure over a wide temperature range. Tensile tests from cryogenic temperature to above 1000°C in combination with creep test have been conducted and results compared to existing similar materials. Results indicate the new alloy to exhibit excellent mechanical performance, particularly at evaluated temperatures, compared to AM processed non-ODS and Inconel 718 alloys. The article is overall written clearly and the topic timely. Despite these positive aspects, the main issue with this manuscript is that the authors simply report results – that are in various places incomplete – without characterizing deformation mechanisms and investigating the microstructure relationships with the obtained properties (the fact that the authors state that 'reasons for the performance are still being explored' makes it according to the npg publication guidelines of their various journals most suitable for Scientific Reports!). Also, statements such as ODS alloying can increase an alloy's high temperature mechanical and oxidation properties – independent of the alloy – kind of disqualifies the manuscript from publication in such high impact journal not only because this is known but this also brings to question what advantage the compositionally complex alloy design approach has? If additive manufacturing should be the unique part of the manuscript then the problem arises that there are many details about the processing missing – essentially all of them. How, when, and where was printing optimized? What parameters were finally used for printing? Speculations about solute segregation of carbon, extended SF node configurations, and STFs towards the end of the manuscript without links to prior sections in the paper or mechanisms are also not helpful. Finally, comparisons to other materials seem to be made somewhat carefully but materials such as Ni-based CMSX-4 are still missing, and it is unclear why certain comparisons are made of samples that were tested in horizontal/vertical testing orientation with respect to the build direction and others are not or why at all the new alloy has been hiped if the properties are in many aspects worse than the non-hipped alloy. Finally, what's the point of naming the material GRX-810 if nothing of the alloy points towards this temperature being specifically important?!

Further issues include:

1. While the comparison with NiCoCr, NiCoCr-ODS appears to make sense it is unclear why ODS-ReB was used. Furthermore, comparisons with Inconel 718 were made only for the creep tests and oxidation results but not for tensile experiments (at elevated temperatures). Why?
2. "Pg 4 Line 87: Enable MC carbide formation along grain boundaries which are stable above 1200 °C." Why did the authors pick that temperature limit for their simulations?
3. Was the optical density measurement carried out on the single big image shown in SI Fig. 2 or on multiple small regions with the area shown in that figure. This makes a difference in terms of density characterization. Also, relative density measurements should additionally be made using Archimedes principle.
4. Why is Co not shown in STEM-EDS image in Fig.2?
5. Tensile Tests:
 - a. How many tensile tests were conducted per condition?
 - b. For the HIP GRX-810 condition, two orientations were tested. Why haven't the authors carried out similar tests in the as-built condition and the other materials?
 - c. Fig. 4b clearly shows that the orientation of the samples impact the measured mechanical performance, yet they simply ignore this detail throughout the manuscript without even mentioning in what orientation the other samples that are shown in Fig. 4a,c,d were tested.
 - d. Results from Supplementary Table 1 and 2 show that with increasing temperature the ductility of the material increases at 871 °C and then decreases at 1093 °C. Why?
 - e. In Fig. 4d it is unclear which data points have actually been taken and what part of the curve is fitted.
 - f. Pg 10 Line 193: "This result highlights the anisotropy present in the AM samples which can't be recrystallized through conventional means, but also suggests the print direction is a weaker orientation for these ODS materials." – Do the authors mean the HIP treatment as "conventional means"? When the authors say "the print direction is a weaker orientation" do they mean weaker in-terms of strength or ductility? This is very vague language.
 - g. Tensile tests of ODS-ReB samples at 1093 °C show the HIPed sample to have low ductility compared to the as-built sample. One would expect the opposite. So what is causing this behavior. Again, there is a significant lack of discussion about mechanisms in this manuscript.
6. Creep tests:
 - a. It is somewhat difficult understand these results given that various loading conditions have been used: 14 MPa (AM 625 HIP), 20 MPa (Fig.5a,b), 31 MPa (Fig.5c,d), 41MPa (Supplementary Fig.5). While most of them are OK, individual sample conditions are missing at every stress level making the results difficult to compare.
 - b. Fig.5 caption: "The HIP GRX-810 curve at 20MPa is still in testing (though in tertiary creep) while the as-built test was terminated at 1 % strain." – What do the authors mean by "terminated"? did they stopped the test (and if so, why?) or did the sample fail?! Also, what testing environment was used?
 - c. Fig.5c – As-built NiCoCr, ODS-ReB samples performed better than their HIPed counterparts. This is a very strange result - again, why did this happen and what's the mechanisms behind this? HIPed samples often have lower porosity which should positively impact the results and lead to better creep performance!
 - d. HIPed GRX-810 creep test at 31MPa was omitted. Why?
 - e. It is also unclear whether the creep test results shown in Fig.7 for GRX-810 samples were from the as-built or HIPed material?
7. Oxidation results:
 - a. Why were NiCoCr, NiCoCr-ODS, ODS-ReB not used for this characterization?
 - b. Fig.6c caption: "(c) Optical images of the oxidation samples after 100 hours at 1093°C and 3 hours at 1200°C, where the superalloy 718 sample presented catastrophic oxidation." Only two conditions were mentioned in the caption, what is the third sample shown in Fig.6c?
8. "The slight additions of Re and B to the NiCoCr-ODS sample may have provided slightly higher strengths to the alloy but more testing is needed to confirm this result." – What do the authors mean by "more testing is needed to confirm this result"? Shouldn't this be clear before submission

to Nature?

9. Details in the Mechanical Testing section are missing, e.g., how was strain measured and in which orientation were the samples tested? Furthermore, details of this section are in contrast to the actual tests being conducted. This is confusing.

Again, the authors show impressive results but there is not really a discussion which is currently also not really possible as it seems there has been no characterization of the material after testing so that the manuscript currently primarily shows results and compares them to few other materials.

2nd November 2022

Reviewer comments on the significance and outstanding features of this work

In this manuscript and building upon their previous research [1], Smith et al. present a high-quality report on the microstructure and high temperature performance of the oxide dispersion strengthened, NiCoCr-based, multi-principal element alloy GRX-810. The alloy was designed using an integrated computational materials engineering approach, which facilitates rapid alloy development. The compositional design requirements were such that a i) face-centred solid solution should be maintained, ii) the freezing range of the alloy should be 100°C or less to improve processability via additive manufacturing, iii) metal carbides can form along grain boundaries, which are stable above 1200°C and that iv) solid-solution strengthening should be maximised. The ICME approach used in this study falls outside my area of expertise, so I will discuss the following outstanding features of this original piece of work.

The quality of and detail provided in the electron microscopy is truly impressive; there is clear evidence of a uniform dispersion of nanoscale Yttria particles in the matrix and TEM studies showed no evidence of short-range ordering. Achieving a uniform dispersion of oxide particles via Additive Manufacturing without resource intensive steps is a worthy achievement and this is highlighted in their previous work [1].

The authors present strong evidence that the oxide dispersion, along with solid-solution strengthening and the presence of metal carbides has a remarkable impact on the high temperature creep performance of GRX-810, although the interaction between the various potential strengthening modes is currently subject to further studies. Along with improved high temperature strength and significant resistance to oxidation when compared with additively manufactured Inconel718, the key achievement presented in this work is the high-temperature creep performance of GRX-810. This is illustrated in the form of a stress-rupture diagram (Fig. 7 of the submitted manuscript), where the remarkable increase in the time to rupture of GRX-810 at 1093°C is compared against conventional nickel-base alloys fabricated by Additive Manufacture. From the compelling evidence presented in the manuscript, the creep rupture life of GRX-810 is clearly several orders of magnitude greater than alloys such as Haynes 233, Hastelloy X and Inconel 625. Could the authors please clarify the following however:

- The caption for figure 7 in the manuscript states that the creep performance of GRX-810 is compared against current art alloys used in 3D-printed high-temperature applications. In line 294 of the body however, it suggests the creep data for Hastelloy X and Inconel 625 (Refs. [46-51]) etc. is for the wrought condition. Judging by the publication dates of Refs. [46 – 51], I suspect the latter is the case i.e., the creep data is for wrought alloys, rather than the AM condition.

Due to the exceptional creep performance of GRX-810 at 1093°C, I was curious to understand how the creep data presented in the manuscript compared with the creep performance of both conventional and additively manufactured Ni-based superalloys operating in the lower temperature, higher stress regime. To do this, I calculated the Larson-Miller Parameter (P) for the GRX-810 rupture time data¹ using Equation 1 and obtained values of $P = 29.3, 30.4$ and 32.0×10^3 (3 s.f.) for applied stresses of 51, 41 and 31 MPa, respectively (calculations are provided in the appendix):

$$P = T[20 + \log_{10}(t)] \quad \text{Eq. 1}$$

Where T = Temperature (K) and t = time to rupture in hours.

These values, along with Larson-Miller Parameter data for conventionally processed Ni-based superalloys from page 18 in Reed [2] and Additively Manufactured examples from Tang *et al.* [3] and Körner *et al.* [4] are presented in Figure 1.

¹ The stress and time to rupture data for GRX-810 were extracted from Figure 7 in the manuscript using the App “Web Plot Digitizer”, so the values are accurate, but somewhat imprecise. See comments on the use of axis tick marks in the section “Reviewer comments on data presentation”.

Although care should be taken when extrapolating between the low temperature, high stress regime and high temperature, low stress regime in a Larson-Miller plot [5], GRX-810 appears to perform equally as well and if not better than Additively Manufactured ABD-900 [3] and CM247 [3]. One could also argue that creep performance of GRX-810 is comparable to that of the superalloys CMSX-4 and RR3000 in the low stress, high temperature regime if the downward slope of the Larson-Miller curve for these two alloys continues its trajectory. Interestingly, the Larson-Miller curve for GRX-810 appears to have a similar form to MA754, an oxide-dispersion strengthened Ni-alloy processed by conventional means. i.e. improved performance at higher stresses and temperature to the detriment of its high stress, low temperature properties.

- The authors may wish to present their creep data for GRX0-810 in the form of a Larson-Miller diagram similar to that shown in Figure 1. This is only a suggestion however, and it is not a requested revision to this manuscript.

As stated previously, care should be taken when comparing using Larson-Miller diagrams to compare the creep performance between the low temperature, high stress and the high temperature, low stress regimes, as higher homologous testing temperatures could lead to changes in metallurgical structure [5] and the dominant mechanism of creep deformation e.g. a change from dislocation creep to diffusional flow [6]. As the authors state in their manuscript, they intend to fully investigate the mechanisms governing creep deformation and strengthening at 1093°C. This is an exciting avenue of exploration and I feel it would be very interesting to assess the creep performance (and the associated mechanisms of deformation) of GRX-810 at higher stresses and lower homologous temperatures in future studies.

Figure 1. A Larson-Miller diagram indicating the creep performance of GRX-810 relative to the performance of conventionally processed Ni-alloys reported by Reed [2], additively manufactured Ni-alloys reported by Tang [3] and Körner [4], and the performance of the Ni-alloys reported in Refs. [46 – 51] in the submitted manuscript to *Nature*.

Reviewer comments on the experimental approach

A comprehensive description of the experimental and modelling methods has been provided by the authors, including details of the microstructural characterisation, oxidation testing and mechanical testing methods. The reporting of the methodology is sufficiently detailed and transparent such that other researchers could largely reproduce the results. I'm unable to comment on the density functional theory calculations, as this falls outside my area of expertise.

Details of the additive manufacturing methods are largely provided, including a statement that the specimens were built using an EOS machine platform with a beam radius (r_B) of 20 microns using feedstock powder sourced from Praxair Inc. No further details pertaining to the additive manufacturing processing parameters are given; for example, the laser beam power and velocity, layer height and hatch spacing. The reviewer appreciates that these parameters are likely to be sensitive and the authors may not wish to disclose this information at present. Methods to normalise the processing parameters do exist however, see for example Fig. S5 in the supplementary data section of Wang et al. [7]. Normalisation of the laser power and velocity would require estimates of the thermo-physical properties of GRX-810 such as the melting temperature, density and thermal conductivity, and the approach is described in the methods section of Wang et al. [7]. The authors may therefore wish to consider presenting their processing parameter data in dimensionless or normalised form.

Reviewer comments on data presentation

Figures are generally of a high quality throughout the manuscript, although there are a few points of clarification and suggested improvements I would like to make.

- Do the axes of the ternary phase diagram in Figure 1 b) represent the weight fraction, atomic fraction or mole fraction of each alloy component?
- The authors may wish to double check the scale bar in Figure 2 a). As presented, I interpret the scale bar to be 500 microns, which suggests the Yttria particles are of a 20 to 50-micron length scale. Likewise, in line 153 it is stated that there is no ordering at the 500-micron length scale. Should this not be 500 nm as the high magnification TEM image in Figure 3 b) indicated the dispersed oxides are ~20 to 50 nm in length, which seems more accurate?
- Could the authors please clarify whether the stress-strain curves in Figure 4 of the manuscript refer to true or engineering stress and strain. From the shape of the flow data, I presume it's the latter.
- In figure 5 b), the creep curve for the as-built GRX-810 sample is difficult to differentiate from the HIP specimen. The creep test for the as-built specimen was terminated at a strain of 1%, so the graph would benefit from marker/symbol indicating the rupture time.
- Could the authors please define what the error bars in Figure 6 a) correspond to in the figure caption? Is this the measurement uncertainty, the range in any replicate results or the standard error of the mean etc.? Likewise, please define what the error bars correspond to in Figure 6 of the supplementary information.
- The graphs in the manuscript and supplementary data showing the tensile data, the creep curves, the oxidation mass change, and stress rupture data would also benefit from axis tick-marks.

Concluding remarks

Overall, this is an excellent paper that presents results that are of immediate interest to both scientists and engineers working in the field of metallurgy and materials science. References are appropriate throughout and the authors may also wish to refer to the work of Tang *et al.* [3] in their introduction as this is another good example of a computational approach to the design of high temperature alloys Additive Manufacturing applications.

The data presented in this manuscript are reliable, and the interpretations and conclusions are sound. To the best of my knowledge, the manuscript does not contain any inflammatory material and I have no concerns that this manuscript has any impact on Springer Nature's commitment to equality, diversity or inclusion.

References

- [1] Smith, T.M., Thompson, A.C., Gabb, T.P. *et al.* Efficient production of a high-performance dispersion strengthened, multi-principal element alloy. *Sci Rep* **10**, 9663 (2020).
- [2] Reed, R. C., *The Superalloys: Fundamentals and Applications*. Cambridge University Press (2006).
- [3] Tang, Y. T., Panwisawas, P., Ghossoub, J. N., *et al.* Alloys-by-design: Application to new superalloys for additive manufacturing. *Acta Materialia* **202**, 417-436 (2021).
- [4] Körner, C., Ramsperger, M., Meid, C., *et al.* Microstructure and Mechanical Properties of CMSX-4 Single Crystals Prepared by Additive Manufacturing. *Metallurgical and Materials Transactions A* **49**, 3781-3792 (2018).
- [5] Dieter, G. E., *Mechanical Metallurgy* (S. I. Metric Edition). McGraw-Hill (1988).
- [6] Ashby, M. F., A first report of deformation-mechanism maps. *Acta Metallurgica* **20**, 887-897 (1972).
- [7] Wang, Y., Voisin, T., McKeown, J. *et al.* Additively manufactured hierarchical stainless steels with high strength and ductility. *Nature Materials* **17**, 63–71 (2018).

Appendix: Larson-Miller Parameter Calculations for GRX-810

The Larson-Miller parameter is calculated using Eq. A1:

$$P = T[C + \log_{10}(t)] \quad \text{Eq. A1}$$

Where C is taken to be 20 as per Ref [2] for the purpose of this brief analysis. The stress-rupture data for GRX-810 and other alloys were extracted using the App “Web plot digitizer”

1. For GRX-810:

T (°C)	T (K)	Stress (MPa)	Rupture time (h)	P	P (x10 ³)
1093	1366	51.3	27.2	29279	29.3
1093	1366	41.3	184.9	30417	30.4
1093	1366	31.0	2527.6	31968	32.0

2. For other high-temperature alloys (Refs. [46-51] in the submitted manuscript):

T (°C)	T (K)	Stress (MPa)	Rupture time (h)	P	P (x10 ³)
1093	1366	41.2	3.4	28043	28.0
1093	1366	31.1	12.6	28822	28.8
1093	1366	31.1	13.4	28859	28.9
1093	1366	20.7	90.1	29990	30.0
1093	1366	30.9	1.3	27468	27.5
1093	1366	30.9	1.4	27518	27.5
1093	1366	27.5	1.6	27598	27.6
1093	1366	30.9	2.7	27920	27.9
1093	1366	30.9	3.6	28074	28.1
1093	1366	22.6	10.0	28686	28.7
1093	1366	20.8	6.9	28463	28.5
1093	1366	18.8	10.1	28692	28.7
1093	1366	20.6	13.1	28847	28.8
1093	1366	13.7	15.5	28946	28.9
1093	1366	10.3	93.0	30009	30.0
1093	1366	14.4	103.2	30071	30.1
1093	1366	15.3	101.0	30058	30.1

3. For Additively Manufactured CMSX-4 [4]:

T (°C)	T (K)	Stress (MPa)	Rupture time (h)	P	P (x10 ³)
1050	1323	160	204.3	29516	29.5
850	1123	600	307.7	25254	25.3

4. For all other data presented in Figure 1:

Data points for P versus Stress from the Larson-Miller curves shown in Figure 1.16 (p. 18) of Reed [2] were extracted using web plot digitizer.

Figure 1.16 from Reed [2] is provided below:

Fig. 1.16. Values of the Larson–Miller parameter, P , for a number of high-temperature materials. The horizontal and near-vertical lines have spacings equivalent to a factor of 2 change in creep life and 200 °C temperature capability, respectively; the materials which display the best high-temperature performance lie towards the top right of the diagram. Adapted from a graph provided by Dan Miracle.

Notes:

Nature ask reviewers the following questions, to provide an assessment of the various aspects of a manuscript. Please find my responses below:

- Key results: Please summarise what you consider to be the outstanding features of the work. **YES. See comments in report.**
- Validity: Does the manuscript have flaws which should prohibit its publication? If so, please provide details. **NO**
- Originality and significance: If the conclusions are not original, please provide relevant references. On a more subjective note, do you feel that the results presented are of immediate interest to many people in your own discipline, and/or to people from several disciplines? **YES.**
- Data & methodology: Please comment on the validity of the approach, quality of the data and quality of presentation. Please note that we expect our reviewers to review all data, including any extended data and supplementary information. Is the reporting of data and methodology sufficiently detailed and transparent to enable reproducing the results? **YES. See comments in report.**
- Appropriate use of statistics and treatment of uncertainties: All error bars should be defined in the corresponding figure legends; please comment if that's not the case. Please include in your report a specific comment on the appropriateness of any statistical tests, and the accuracy of the description of any error bars and probability values **See comments in report.**
- Conclusions: Do you find that the conclusions and data interpretation are robust, valid and reliable? **YES.**
- Suggested improvements: Please list additional experiments or data that could help strengthening the work in a revision. **Comments made on data presentation.**
- References: Does this manuscript reference previous literature appropriately? If not, what references should be included or excluded? **YES. See comments in concluding remarks.**
- Clarity and context: Is the abstract clear, accessible? Are abstract, introduction and conclusions appropriate? **YES.**
- Inflammatory material: Does the manuscript contain any language that is inappropriate or potentially libelous? **NO.**
- Springer Nature is committed to diversity, equity and inclusion; please raise any concerns that may in your view have an impact on this commitment. **No concerns.**
- Please indicate any particular part of the manuscript, data, or analyses that you feel is outside the scope of your expertise, or that you were unable to assess fully **YES. ICME approach including the DFT Calculations.**
- Please address any other specific question asked by the editor via email.

Author Rebuttals to Initial Comments:

Reviewer 1:

- **maybe state here to WHICH Ni alloy / alloy group this comparison refers to (because the variety in properties of such alloys is quite large; which material is the closest competitor etc.)?**
 - o Both in the abstract and introduction we now mention that our comparisons are to polycrystalline wrought Ni-base superalloys that are currently being used for high temperature AM components. We have had extensive discussions with U.S. industry and government partners, and this would be the alloy systems that GRX-810 would most likely compete with and replace.

- **1. Using / exploiting the in-situ reaction of oxygen with specific alloying elements for material design in additive manufacturing was shown a few times before in literature and hence it should be made clear what the specific novelty here is compared to the state of the art before.**
 - o Our process of producing ODS alloys does not use In-situ alloying during the AM process. Instead, we coat our base metal feedstock with nano-scale oxides through a high energy mixing step. More description of this process has been included in the introduction and methods sections.
 - o “Investigations of oxide dispersion strengthened multi-principle element alloys have revealed improved high temperature properties (strength and creep)²⁷ and irradiation properties²⁸. Similarly, multiple recent studies have successfully produced ODS alloys through L-PBF through a variety of techniques^{27,29,30}. These techniques have relied upon mechanical alloying^{27,30}, In-situ alloying²⁹, or chemical reactions³¹ to introduce and incorporate oxides into the 3D printed matrix. However, all these processes introduce complexity and repeatability issues when trying to produce similar material through different AM methods or machines. Recent work by Smith *et al.* produced oxide dispersion strengthened NiCoCr through laser powder bed fusion (L-PBF) where nano-scale Y₂O₃ nano-particles were coated onto the NiCoCr metal powder feedstock through a high energy mixing process that does not require any binders, fluids, or chemical reactions. This process did not deform or impact the powder spherical morphology which is important for high quality AM components. Using this approach they produced an ODS alloy that provided a 35% increase in tensile strength and 3x improvement in ductility at 1093°C compared to its non-ODS counterpart³².”
 - o

- **2. The sentence ‘These results represent a paradigm shift in alloy development, where ICME combined with AM can be leveraged to accelerate the development of revolutionary materials ‘ is maybe a bit too bombastic (not really necessary here?) – if the results are so strong as they are, better be more specific on the exact scientific / engin. novelty items; better work out / state what is the more specific innovation – how can I translate to other creep alloys too etc – maybe be a bit more specific here. What is the innovation – is it the fine dispersion? Is it to do that in-situ during AM; is it to control the O2 partial pressure during AM? Better be more**

specific here on the innovative and inventory height – this is more helpful than a less specific ‘sales pitch’.

- All references to ICME have been replaced with more descriptive terms. This last sentence has also been changed to the following which we feel is less bombastic.
 - “These results showcase how future alloy development which leverages model-driven alloy design combined with AM can accelerate the discovery of revolutionary materials.”
- **3. The inventory advancement of the topic / field could be highlighted a bit more convincingly: A few papers were published before on the use of Y-based oxide dispersions in such materials, also with high radiation resistance etc.; Example: High radiation tolerance of an ultrastrong nanostructured NiCoCr alloy with stable dispersed nanooxides and fine grain structure, by C Lu et al. In the Journal of Nuclear Materials showed in 2021 very good behaviour of nanostructured NiCoCr medium entropy alloys enhanced by Y-Hf-O nanooxides etc – so, similar as the comment above: better describe a bit more clearly what the actual item(s) of highest progress is, maybe the combination of AM and creep properties etc ?**
 - We agree and have included the new paragraph in the introduction that was presented in our response to the first concern. We have also included the paper from C Lu in the introduction section.
- **4. Readers might not know what ICME is. Recall this is for general readers. Is the term here really required or rather distracting?**
 - This term is no longer used in the paper.
- **5. In the introduction, the authors report in general terms on the particularly favorable deformation mechanisms in the family of Cantor alloys and related materials (strength / ductility trade off etc.)- but it seems that these properties are not quite so relevant to the alloys discussed here, since these features are room-temperature properties? But this paper is about creep? Maybe check storyline in the intro?**
 - We feel the additions/changes made to the introduction section better introduce this work and the novelty of it.
- **6. Not good to use s.t. like ‘GRX-810 in a paper for a general science readership; readers do not know what that is / means. Better pick more general title.**
 - We would really prefer to keep this title but are willing to change it if the Editors agree “GRX-810” should be removed.
- **7. The naming convention ‘This new ODS alloy 20 called GRX-810 (Glenn Research Center Extreme Temperature above 810 °C) is maybe a bit too specific for an abstract of such a journal.**
 - We have removed this sentence from the abstract of the paper.
- **8. Details missing for refs 39, 46, 47, 50 etc. better check all refs.**
 - We have added more details to these references.
- **9. Ref 54: the name’s not ‘Schrodinger ‘...**
 - The reference now states “schrödinger” instead.

- **10. Ref 12 was not the first paper to show that the strength-ductility trade-off can be overcome because of atomic-scale deformation mechanisms. This was reported in earlier papers before for these alloys.**
 - o We now cite: Gali, A. & George, E. P. Tensile properties of high- and medium-entropy alloys. *Intermetallics* **39**, 74–78 (2013).

- **11. The use and beneficial effects of C and other interstitials on HEAs was show in several papers before the here quoted ref 22 – maybe best double check to use the original references also? (I know, not easy in this fast-moving field).**
 - o We now also cite: Wu, Z., Parish, C. M. & Bei, H. Nano-twin mediated plasticity in carbon-containing FeNiCoCrMn high entropy alloys. *J. Alloys Compd.* **647**, 815–822 (2015).

- **12. The authors use the term ‘ICME approach and it seems here that they essentially mean by that that they did phase diagram calculations prior to synthesis, hence, maybe just better state exactly that? The ICME terminology for doing s.t. that is well established is maybe a bit distracting? I would tend to be much more exact and specific in the method / design description and avoid any buzz terminology that is probably unsuited for general readers to follow? The results section is much more clear here, so maybe consider rewriting the intro section scientifically a bit more specific and maybe state ,thermodynamics calculations or so instead of ‘ICME’?**
 - o We agree and again ICME has been completed removed from the paper. We now state in the methods section more clearly what was done.
 - o “Thermodynamic modeling (CALPHAD) was employed to produce a superior composition using equiatomic NiCoCr as a foundation^{57,58}” and note thermocalc which is what we used in the methods section.
 - o “Simulations were performed using Thermo-Calc version 2020b with the Ni alloy database TCNI8. Upwards of 10^7 equilibrium calculations were performed across composition and temperature space.”

- **13. For example they write ‘...an ICME approach was employed to design a NiCoCr system for high temperature applications using AM for complex components. This effort resulted in a new composition ...’ What exactly was done, which steps, which simulation etc. ? The term ICME does not really say anything about the science that was done? I guess it was a phase diagram and a Scheil simulation or so using Thermocalc and Dictra or so ? Better be more specific here in such paragraphs?**
 - o We feel this has been remedied by our responses to previous concerns.

- **14. Statement ‘slight additions of Re and B ‘...? better be more specific – these dopants are quite essential – which range was explored – these are essential information items for understanding the design process. Also, add a Fig reference here. Would be nice to have a bit more specific details in some of these statements.**
 - o We specify the additions in the paper now.

- “[NiCoCr, NiCoCr-ODS, NiCoCr-ODS with slight additions of Re (1.5 wt.%) and B (0.03 wt.%) (ODS-ReB)].”
 - We have also included a table in the supplementary that states the alloy names and corresponding compositions for all the tested and referenced alloys in the paper.
- **15. Statement ‘This study confirms the maturity of both ICME driven alloy design... ‘ Not sure – it was neither explained what ICME is nor was stated what exactly was done ... ICME is more a very general buzzword but the authors need to specifically state how the design was done, which TD / kin. calculations were done etc. using WHICH database etc, screening which range of interstitials etc. – all this info must be briefly added here, as just stating ‘ICME’ did it all is not informative.**
- Again, we feel that this has now been addressed.
- **16. Term ‘alloy “trade space” is unclear.**
- Trade space has been changed to “compositional space”
- **17. Paragraph on DFT and ff: ‘a full overview using density functional theory (DFT) calculations was performed... ‘: be more specific here: were these ground state simulations of free energy simulations (for phases, SFEs etc.)? (a) If the latter is true, which entropy models were used ? (b) if the former is true, the 0K results must be more critically discussed w.r.t. relevance. Fig. 1 seems to suggest we look at 0K data from DFT? This point in the overall design approach is not quite so clear as the authors develop a HIGH-temperature alloy, where entropy is surely essential? Hence, what is the point of discussing 0K results here and what is the expected error / effect for the here targeted high temperature case? Needs a bit deeper discussion I suppose**
- We performed this analysis at 0K, which is now stated in the methods section. We understand that this phase diagram is probably not an accurate description of the most likely stable phases at high temperature, but we feel there is a lot of valuable information in these calculations moving forward. By comparing how far we can push the NiCoCr system into the calculated lower energy HCP phase while maintaining a FCC phase (in practice) we believe we will both improve the high temperature and, more significantly, the low temperature properties of the alloy. We chose to use “extreme environment” in the title and not “high temperature environment” because this alloy performs so well at both temperature extremes. One reason for this is the predicted HCP phase formation that occurs during deformation at cryogenic temperatures which is somewhat highlighted in this figure. The below text has been added to the methods section.
 - “For future guidance in NiCoCr-based alloy development, the results from Extended Data Fig. 9 are organized into a predicted ternary phase diagram at 0 Kelvin in Fig. 1(b). Though this phase diagram may not represent the stable phases at high temperatures due to entropy, these calculations have important implications in the properties of the NiCoCr system at cryogenic temperatures. Recent papers have found excellent mechanical properties in NiCoCr-based medium entropy alloys at these low temperatures where the phase transformation from FCC to HCP during deformation is

the principle contributing factor^{61,62}. Therefore, these low temperature properties may be further improved by moving the composition of NiCoCr into the more stable HCP phase regime, while maintain an FCC phase. Future work is planned to explore this possibility.”

- **18. Powder production: was the material pre-alloyed and WHICH composition was pre-alloyed ? I assume the pre-alloyed powder were low in O and the O came into play during AM? Maybe give more info here on the intermediate states?**
 - o All the base metal powder was purchased pre-alloyed. This is now specifically clarified in the methods section.

- **19. The stacking fault discussion / info regarding Fig 3 is from a room temperature image – what is the estimated role / existence of the SFs and SF tetrahedra at operation temperature / high temperatures ? Do these features matter for creep or are they mere ambient temp features ?**
 - o We believe these features are notable which is why we briefly discussed them. It is our current thought that these features can be additional obstacles for dislocation motion during creep at high temperatures. However, we have now added more characterization and analysis in the manuscript where we conclude that the MC carbides and solute segregation around grain boundaries are the main reasons for the exceptional creep performance in GRX-810.

- **20. Fig. 4: better to add reference data from a few established (commercial) Ni base superalloys into the same diagram ? Non-expert readers might find it otherwise hard to see the differences to established alloys in this field ? (similar as nicely summarized in Fig 7 – this seems to be the real key fig of the paper ? maybe add some / a few of these ref data also to Fig 4 ?)**
 - o Again, we are not really sure what the role is for the SF tetrahedra but we believe they play a role as additional dispersoids inhibiting dislocation motion as is mentioned in the study. The revised manuscript puts more focus on the grain boundary strength of GRX-810 in explaining the creep properties though.
 - o We have included the below graph to better compare tensile strengths between GRX-810 and other superalloys. Both the wrought Haynes 230 and the non-ODS GRX-810 tests were added to this study during the revision.

- **21. Some abbreviations and material names used in the legends etc are unknown to general readers, such as e g H230 etc – all should be clearly given / explained etc.**
 - o H230 has been replaced to Haynes 230 in all figures where it occurred.
- **22. In Fig 7 (and all figs): pls make sure that ALL abbreviations / alloy names etc. are explained somewhere in the paper / overview table etc. / give compositions etc. – that would be really helpful.**
 - o The following table is now included in the supplementary

Alloy	Ni	Cr	Co	Mo	W	Nb	Ta	Al	Ti	Re	Fe	B	C	Y ₂ O ₃
Hastelloy X	41.9	21	1	9	1	0	0	7	1	0	18	0	0.1	0
Inconel 718	52.56	19	1	3	0	5	0	0.5	0.9	0	18	0	0.04	0
Inconel 617	55.13	22	12.5	9	0	0	0	1	0.3	0	0	0	0.07	0
Haynes 230	61.6	22	0	2	14	0	0	0.3	0	0	0	0	0.1	0
Haynes 188	22	22	42	0	14	0	0	0	0	0	0	0	0	0
Inconel 625	62.95	21.5	0	9	0	3.6	0	0.2	0.2	0	2.5	0	0.05	0
Haynes 233	48	19	19	7.5	0.3	0	0.5	3.3	0.5	0	0	0.004	0.1	0
NiCoCr-ODS	33.7	30.7	34.7	0	0	0	0	0	0	0	0	0	0	1
NiCoCr	34	31	35	0	0	0	0	0	0	0	0	0	0	0
ODS-ReB	33.0	30.0	34.0	0	0	0	0	0	0	1.5	0	0.003	0	1
GRX-810	31.2	33	29	0	3	0.75	0	0.3	0.25	1.5	0	0	0.05	1
GRX-810 non-ODS	31.5	33.3	29.3	0	3	0.75	0	0.3	0.25	1.5	0	0	0.05	0

Reviewer 2:

- **The caption for figure 7 in the manuscript states that the creep performance of GRX-810 is compared against current art alloys used in 3D-printed high-temperature applications. In line 294 of the body however, it suggests the creep data for Hastelloy X and Inconel 625 (Refs. [46- 51]) etc. is for the wrought condition. Judging by the publication dates of Refs. [46 – 51], I suspect the latter is the case i.e., the creep data is for wrought alloys, rather than the AM condition.**
 - o You are correct that this data is from wrought versions of common alloys used for AM. We more clearly specify this in the description of Figure 7 (now 5).
 - o **“Fig. 5: Creep rupture life of GRX-810 compared to current SOA superalloys^{44,51–55}.** Scatter plot of superalloy creep rupture life at 1093°C. GRX-810 presents superior creep

properties compared to wrought alloys currently used in 3D printed high temperature applications.

- **The authors may wish to present their creep data for GRX0-810 in the form of a Larson-Miller diagram similar to that shown in Figure 1. This is only a suggestion however, and it is not a requested revision to this manuscript.**
 - o We seriously considered using the approach to compare creep strength of GRX-810 to other alloys. We also greatly appreciate the analysis provided in your review. Still, as was the case before we first submitted the paper, we are uncomfortable using the approach to compare alloy properties. First, we feel that to have truly accurate Larson-Miller parameters we need more data from different temperatures and stresses. Secondly, the comparison of this alloy to other directionally recrystallized/solidified or single crystal alloys brings up more questions and discussion that falls outside the scope of this specific paper. As we plan to write and publish more on this alloy we will be sure to incorporate Larson-Miller parameters.

- **Details of the additive manufacturing methods are largely provided, including a statement that the specimens were built using an EOS machine platform with a beam radius (rB) of 20 microns using feedstock powder sourced from Praxair Inc. No further details pertaining to the additive manufacturing processing parameters are given; for example, the laser beam power and velocity, layer height and hatch spacing. The reviewer appreciates that these parameters are likely to be sensitive and the authors may not wish to disclose this information at present. Methods to normalise the processing parameters do exist however, see for example Fig. S5 in the supplementary data section of Wang et al. [7]. Normalisation of the laser power and velocity would require estimates of the thermo-physical properties of GRX-810 such as the melting temperature, density and thermal conductivity, and the approach is described in the methods section of Wang et al. [7]. The authors may therefore wish to consider presenting their processing parameter data in dimensionless or normalised form.**
 - o The reasons that print parameters have been excluded from this publication is because the U.S. Government has deemed the EAR restricted, and we are not allowed to do so. We spent a significant amount of time discussing with our export control managers to see what else we could include in the paper. We now include the energy density of the print parameters which incorporates laser power, speed, hatch spacing, etc.
 - o “For GRX-810 builds on the EOS M280, the best densities were achieved with a laser energy density of 90-110 J/mm³”
 - o We understand that it would be useful to include more information but someone who is has experience with AM will get at least a sense of the parameters used through these energy density ranges.

- **Do the axes of the ternary phase diagram in Figure 1 b) represent the weight fraction, atomic fraction or mole fraction of each alloy component?**
 - o Atomic fraction - this is now specified in the figure description.

- The authors may wish to double check the scale bar in Figure 2 a). As presented, I interpret the scale bar to be 500 microns, which suggests the Ytria particles are of a 20 to 50-micron length scale. Likewise, in line 153 it is stated that there is no ordering at the 500-micron length scale. Should this not be 500 nm as the high magnification TEM image in Figure 3 b) indicated the dispersed oxides are ~20 to 50 nm in length, which seems more accurate?
 - o Great Catch! This scale has now been changed to 500nm which is what it should have showed previously.
- Could the authors please clarify whether the stress-strain curves in Figure 4 of the manuscript refer to true or engineering stress and strain. From the shape of the flow data, I presume it's the latter.
 - o The curves have all been specified as engineering stress-strain curves in the figure captions.
- In figure 5 b), the creep curve for the as-built GRX-810 sample is difficult to differentiate from the HIP specimen. The creep test for the as-built specimen was terminated at a strain of 1%, so the graph would benefit from marker/symbol indicating the rupture time.
 - o This graph now has a description and arrow pointing to when this test was terminated.

- Could the authors please define what the error bars in Figure 6 a) correspond to in the figure caption? Is this the measurement uncertainty, the range in any replicate results or the standard error of the mean etc.? Likewise, please define what the error bars correspond to in Figure 6 of the supplementary information.
 - o These are bars correspond to 1 standard deviation and this is now described in the figure description.
- The graphs in the manuscript and supplementary data showing the tensile data, the creep curves, the oxidation mass change, and stress rupture data would also benefit from axis tickmarks.
 - o Tick marks have been added to all the graphs.

Reviewer 3:

- **Despite these positive aspects, the main issue with this manuscript is that the authors simply report results – that are in various places incomplete – without characterizing deformation mechanisms and investigating the microstructure relationships with the obtained properties (the fact that the authors state that ‘reasons for the performance are still being explored’ makes it according to the npg publication guidelines of their various journals most suitable for Scientific Reports!).**
 - We have added significantly more detail and characterization describing the mechanism behind the great creep performance of GRX-810. The below description has been added to the paper. We performed a new tensile test on non-ODS GRX-810 to help clarify the benefits the MPEA composition provides. We analyzed cross-sections of post-creep samples comparing deformation in ODS-ReB and GRX-810. We have new STEM-EDS evidence revealing W, Cr, and Re segregation along grain boundaries in GRX-810. We also briefly explored the dislocation structure in post-creep GRX-810 samples. This new evidence was used to write the mechanistic reason for GRX-810’s performance shown below.
 - “While it is evident that the addition of oxide dispersoids improves both mechanical and oxidation properties, STEM analysis of the ODS-ReB and GRX-810 did not reveal any significant difference in the oxide size or spatial distributions which could be used to explain the differences in creep performance between the two alloys. Therefore, to better explain the creep strength of GRX-810 longitudinal sections taken from two 1093°C air creep tests at 20 MPa were analyzed as shown in extended data Fig. 7.

Extended Data Fig. 7: Optical Analysis of Creep deformation: (a) An optical cross-section of the as-built ODS-ReB sample tested at 1093°C / 20 MPa. (b) creep pores/overload cracking along with a Cr-rich nitride in a high plasticity region of the sample. (c) A representative micrograph of the microstructure in the as-built GRX-810 sample tested at 1093°C / 20 MPa that was terminated at 1% strain after 2800 h. No creep void formation is observed but the presence of Al-rich and Cr-rich nitride phases along grain boundaries can be seen. (d) A region removed from the fracture surface that reveals creep void formation and lack of nitride formation. (e) A representative micrograph of the microstructure from the grip section of the same as-built GRX-810 sample revealing that nitrides did not form in an area under no stress. These results suggest that the nitride phases are creep induced.

While other ODS alloys (e.g. ODS-ReB) failed from a combination of grain boundary creep void coalescence and shear failure, GRX-810 appears to have suppressed these failure mechanisms as no apparent grain boundary voids/defects were observed after much longer test times. One contributing factor is that non-ODS GRX-810 has higher strength than even previous ODS alloys, thus the creep stress is a lower proportion of the flow stress. However, the grain boundary failure modes in other ODS alloys suggest that in GRX-810 the stable MC carbides and solute segregation of W, Cr, and Re along grain boundaries are also contributing to protecting the alloy from grain boundary failure mechanisms. Therefore, though nano-scale oxides provide sufficient strength in the matrix to avoid dislocation motion as shown in Supplementary Fig. 3. the oxides themselves are not enough to prevent grain boundary failure modes as shown in extended data 7.”

- Also, statements such as ODS alloying can increase an alloy’s high temperature mechanical and oxidation properties – independent of the alloy – kind of disqualifies the manuscript from publication in such high impact journal not only because this is known but this also brings to question what advantage the compositionally complex alloy design approach has?

- We now include mechanical test data from as-built GRX-810 without the oxides and wrought Haynes 230 to better understand the effect the MPEA composition has on the overall properties of GRX-810. As you can see the MPEA composition provides similar strength compared to the ODS version but has similar ductility to the non-ODS NiCoCr. This higher strength combined with the microstructure stability provided by the oxides results in the great creep strength of GRX-810.

- If additive manufacturing should be the unique part of the manuscript then the problem arises that there are many details about the processing missing – essentially all of them. How, when, and where was printing optimized? What parameters were finally used for printing?
 - As was mentioned in our response to reviewer 2. The reasons that print parameters have been excluded from this publication is because the U.S. Government has deemed

them EAR restricted, and we are not legally allowed to do so. We spent a significant amount of time discussing with our export control managers to see what else we could include in the paper. We now include the energy density of the print parameters which incorporates laser power, speed, hatch spacing, etc.

- “For GRX-810 builds on the EOS M280, the best densities were achieved with a laser energy density of 90-110 J/mm³”
 - We understand that it would be useful to include more information but someone who is has experience with AM will get at least a sense of the parameters used through these energy density ranges. We would like to add that a high percentage of AM papers do not explicitly reveal the parameters they use.
- **Speculations about solute segregation of carbon, extended SF node configurations, and STFs towards the end of the manuscript without links to prior sections in the paper or mechanisms are also not helpful.**
- We hope the additional data and discussion helps remedy this concern.
- **Finally, comparisons to other materials seem to be made somewhat carefully but materials such as Ni-based CMSX—4 are still missing, and it is unclear why certain comparisons are made of samples that were tested in horizontal/vertical testing orientation with respect to the build direction and others are not or why at all the new alloy has been HIPped if the properties are in many aspects worse than the non-HIPped alloy.**
- We compared this alloy to wrought superalloys that are commonly used for high temperature AM applications. GRX-810 was developed to specifically replace these alloys. CMSX-4 and other single crystal alloys are not 3D printable (realistically anyway) and have significantly more microstructural differences (70% gamma prime volume fraction compared to a solid solution alloy) that would need to be discussed which we feel falls outside the scope of this study. We also do not have any data on CMSX-4 for the conditions we tested. All the tests were performed in the vertical direction. Only one test was in the horizontal direction. We had to choose which tests to prioritize with the funds available. Therefore, we chose to initially perform the majority of our tests in the print (vertical) direction. This isn't uncommon in exploratory AM studies. We knew that questions would arrive about anisotropy and therefore performed the one tensile test in the horizontal direction as a check to make sure nothing unexpected would arise. We are planning future work where we test a variety of orientations in many different test conditions.
 - We tested HIP'ed versions of each alloy because NASA requires AM components to be HIPped for human rated flight hardware.
- **Finally, what's the point of naming the material GRX-810 if nothing of the alloy points towards this temperature being specifically important?!**
- We have revised figure 1 to show that TCP phases are predicted to become stable at 810°C and therefore we do not advise the alloy to operate at temperatures below that for long periods of time. That's where the number 810 came from for the name.

- **While the comparison with NiCoCr, NiCoCr-ODS appears to make sense it is unclear why ODS-ReB was used. Furthermore, comparisons with Inconel 718 were made only for the creep tests and oxidation results but not for tensile experiments (at elevated temperatures). Why?**
 - o ODS-ReB is just NiCoCr-ODS with 1.5 wt.% Rhenium and .003 wt.% Boron. Thus, making it a closer alloy to GRX-810 than NiCoCr-ODS is. We have added how much Rhenium and Boron is now included in the alloy to help readers better understand what it's actual composition is. The reason we do not have 718 tensile is because we ran out of the limited samples testing creep. Most of our tests on 718 are at much lower temperatures than 1093°C and are not relevant for this study. We have added a new tensile test of wrought Haynes 230 to show a true comparison to GRX-810. Again, non-ODS GRX-810 had significantly higher strength highlighting the benefit this new MPEA compositions is providing.

- **“Pg 4 Line 87: Enable MC carbide formation along grain boundaries which are stable above 1200 °C.” Why did the authors pick that temperature limit for their simulations?**
 - o We set out to make an alloy that could survive operating at temperatures from 1000-1200°C. MC carbides were included to provide grain boundary strength. These carbides would not be beneficial if they dissolved before the alloy reach its operating temperature. Our analysis of failures highlights how the added strength of the MC carbides appears to be a contributing factor towards inhibiting grain boundary void formation and shear cracking.

- **Was the optical density measurement carried out on the single big image shown in SI Fig. 2 or on multiple small regions with the area shown in that figure. This makes a difference in terms of density characterization. Also, relative density measurements should additionally be made using Archimedes principle.**
 - o The optical density measurement was performed over the whole image, which is a composite image of 35 individual micrographs stitched together.

- **Why is Co not shown in STEM-EDS image in Fig.2?**
 - o We did not include Co, due to lack of space and the fact that it is identical to the Ni map and thus would not provide any useful information.

- **How many tensile tests were conducted per condition?**
 - o For this study, 1 test was performed for each condition explored. In some cases we did perform a second test to further validate the result.
 - o We just received 1700 lbs of GRX-810 powder and plan to release a data sheet over the alloy through hundreds of tests when it's completed. Still that work is outside the scope of this manuscript.

- **For the HIP GRX-810 condition, two orientations were tested. Why haven't the authors carried out similar tests in the as-built condition and the other materials?**

- Again, we tested 1 HIP GRX-810 sample to make sure that nothing unexpected occurred, such as a large drop in ductility or strength which would suggest concerning AM defects may still be present.
- **Fig. 4b clearly shows that the orientation of the samples impact the measured mechanical performance, yet they simply ignore this detail throughout the manuscript without even mentioning in what orientation the other samples that are shown in Fig. 4a,c,d were tested.**
 - We now explicitly state that all tests were performed in the vertical direction unless explicitly noted.
- **Results from Supplementary Table 1 and 2 show that with increasing temperature the ductility of the material increases at 871°C and then decreases at 1093 °C. Why?**
 - We are not sure and hope to better understand that change in ductility in future analysis. It is interesting. Still, we feel that answer and analysis is better suited for a different, future manuscript.
- **In Fig. 4d it is unclear which data points have actually been taken and what part of the curve is fitted.**
 - This figure has now been removed and instead we have published the data that was used to make it. See below.
 -
- **Extended Data Table 1: Overview of Tensile Results for As-built and HIP GRX-810.** Tensile results of as-built and HIP GRX-810 at varying temperatures.

Temperature (C)	As-Built Tensile Strength (Mpa)	As-built Yield Strength (Mpa)	As-built Elongation (%)	HIP Tensile Strength (Mpa)	HIP Yield Strength (Mpa)	HIP Elongation (%)
-195.6	1303.1	910.1	39.6	1227.3	723.9	49
21.1	882.5	641.2	33	848.1	515	43
426.7	710.2	527.4	33.3	655	410.2	40
648.9	675.7	479.2	32.1	630.9	368.9	43
871.1	292.3	249.6	56.1	262.7	206.2	62
1000	X	X	X	164.1	161.3	44
1093.3	128.9	127.6	22	119.3	115.8	32

- **Pg 10 Line 193: “This result highlights the anisotropy present in the AM samples which can’t be recrystallized through conventional means, but also suggests the print direction is a weaker orientation for these ODS materials.” – Do the authors mean the HIP treatment as “conventional means”? When the authors say “the print direction is a weaker orientation” do they mean weaker in-terms of strength or ductility? This is very vague language.**
 - This sentence was reworded to be more descriptive.
 - “This result highlights the anisotropy present in the AM samples which can’t be recrystallized through conventional means such as a HIP step, but also suggests the print direction provides less strength than other orientations for these ODS materials.”

- **Tensile tests of ODS-ReB samples at 1093 °C show the HIPed sample to have low ductility compared to the as-built sample. One would expect the opposite. So what is causing this behavior. Again, there is a significant lack of discussion about mechanisms in this manuscript.**
 - o We are not sure why the HIPed ReB-ODS sample exhibited less ductility than the as-built sample. We hope to better understand both this alloy and GRX-810 with future analysis. Still, this paper is primarily focused on GRX-810 and it's microstructure and properties. Even if we did know the mechanism behind this difference, we would probably not include it in this manuscript as it falls outside the scope.
- **It is somewhat difficult understand these results given that various loading conditions have been used: 14 MPa (AM 625 HIP), 20 MPa (Fig.5a,b), 31 MPa (Fig.5c,d), 41MPa (Supplementary Fig.5). While most of them are OK, individual sample conditions are missing at every stress level making the results difficult to compare.**
 - o We hope the reformatted figures make them more easily understood.
- **Fig.5 caption: “The HIP GRX-810 curve at 20MPa is still in testing (though in tertiary creep) while the as-built test was terminated at 1 % strain.” – What do the authors mean by “terminated”? did they stopped the test (and if so, why?) or did the sample fail?! Also, what testing environment was used?**
 - o This graph now has a description and arrow pointing to when this test was terminated.

- o This test did not fail. We ended the test to analyze it's microstructure to better understand why it had such great creep strength. This analysis is now included in Extended data figure 7.
- **Fig.5c – As-built NiCoCr, ODS-ReB samples performed better than their HIPed counterparts. This is a very strange result - again, why did this happen and what's the mechanisms behind this? HIPed samples often have lower porosity which should positively impact the results and lead to better creep performance!**
 - o At the moment, it is our belief that the residual stress and dislocation substructure is providing some benefit to the creep properties of these alloys. All the samples tested had measured densities above 99.9%. Therefore, the improvement the HIP step will provide would be expected to be minimal.

- **HIPed GRX-810 creep test at 31MPa was omitted. Why?**
 - We ran out of the resources to test HIP GRX-810 at this condition. We also now understand that this test is expected to last a few thousand hours making it quite costly to perform with metcut who has performed our creep testing to date. Still, we feel this omission does not impact the quality of our results and/or conclusion.

- **It is also unclear whether the creep test results shown in Fig.7 for GRX-810 samples were from the as-built or HIPed material?**
 - They were for the as-built samples. This is now stated in the figure.

- **Why were NiCoCr, NiCoCr-ODS, ODS-ReB not used for this characterization?**
 - This analysis is outside the scope of this paper. However, we now provide characterization of the creep failures in ODS-ReB in the paper.

- **Fig.6c caption: “(c) Optical images of the oxidation samples after 100 hours at 1093°C and 3 hours at 1200°C, where the superalloy 718 sample presented catastrophic oxidation.” Only two conditions were mentioned in the caption, what is the third sample shown in Fig.6c?**
 - These were the GRX-810 samples at the same moment the 718 sample failed. We know include this in the figure caption to remove any confusion.

- **“The slight additions of Re and B to the NiCoCr-ODS sample may have provided slightly higher strengths to the alloy but more testing is needed to confirm this result.” – What do the authors mean by “more testing is needed to confirm this result”? Shouldn’t this be clear before submission to Nature?**
 - Again, the focus of this paper is not over the ReB-ODS alloy and therefore some of the mechanisms and questions surrounding that alloy are not explored fully. Still, we changed the sentence to the one given below.
 - “The slight additions of Re and B to the NiCoCr-ODS appears to have improved strength slightly.”

- **Details in the Mechanical Testing section are missing, e.g., how was strain measured and in which orientation were the samples tested? Furthermore, details of this section are in contrast to the actual tests being conducted. This is confusing.**
 - We have added new details to the test section that hopefully address any confusion that the readers may have after reading the paper.

- **Again, the authors show impressive results but there is not really a discussion which is currently also not really possible as it seems there has been no characterization of the material after testing so that the manuscript currently primarily shows results and compares them to few other materials.**
 - We hope that the significant additional characterization, analysis, and effort put into this revision has addressed the constructive concerns you put forward. We again thank you for your time in reviewing our manuscript and making it a much more robust and impactful study.

We hope that the newest changes made to the manuscript shows how serious we considered the comments of the three reviewers. We feel that this new manuscript is stronger and more successfully presents our findings. Again, we would like to thank all three reviewers and the editors for their time and constructive feedback which we feel has significantly improved the quality of the paper.

Regards,

Tim Smith

Reviewer Reports on the First Revision:

Referees' comments:

Referee #1 (Remarks to the Author):

I am satisfied with the revision and reply items.

Referee #2 (Remarks to the Author):

18th December 2022

Reviewer comments to the revised manuscript

I would like to thank the authors for the comprehensive revision of their manuscript. I'm satisfied with the authors' responses to my comments and the modifications made. Appreciate the difficulties of sharing the additive manufacturing processing parameters due to export control issues so thank you for supplying the optimised energy density range. I agree that ranking creep performance using the Larson-Miller parameter has its limitations given the small amount of data and I'm looking forward to seeing the results of further creep testing presented in such a way in future publications.

Referee #3 (Remarks to the Author):

In the revised manuscript the authors addressed many points raised in the previous round of reviews and brought their manuscript from a simple report of results closer to something that may be considered for publication – this decision ultimately remains with the editors. While they have added some work to analyze deformation characteristics and add to the mechanistic understanding of the material performance shown in their data, there are several points that remain unclear:

- 1) The authors include a new Extended Data Fig. 7 analysis but do not incorporate this into the discussion of the mechanisms around the performance of the material – which should be the main contribution of the work to the community that warrants publication in Nature (!). The formation of stress induced nitrides, for example, is mentioned in the figure caption but not discussed in the manuscript thereby retaining a sense of 'report character' of their work. Reference to the formation of such nitrides that are common in creep testing of such alloys are missing entirely as details about experimental characterization of their formation are missing; e.g., did they use EDS or any other technique?
- 2) Extended Data Fig. 7b,c shows that Cr-nitrides initiate creep void formation eventually leading to failure. However, such stress induced nitride formation was suppressed in the GRX-810 alloys (Extended Data Fig. 7d,e) even after 2800 h under the same stress experienced by ODS-ReB (Extended Data Fig. 7b,c, which failed at 9 h). This could be because of other alloying element playing a role in suppressing such nitride formation in GRX-810 leading to the superior creep performance. What's the authors take on this? Again, this might be essential to the manuscript and is yet not discussed.
- 3) Not being able to provide details about printing the material is an issues but we understand the importance of government control restrictions. Testing material in only one orientation or only conducting a single test in the second orientation is an issue and somewhat sloppy though – particularly as it doesn't allow for a fair and statistically relevant comparison. Saying that this is

common in the AM community may be true yet a problem with that community that could be 'fixed' by providing a complete set of data published in a high-profile journal like Nature. Saying that 1700 lbs of material have been received and results from the tests will be published later are weak arguments.

4) It is somewhat discouraging that the authors state multiple times that they do not understand a certain materials performance, e.g., the decrease in ductility above 871C, and only mention in the response letter that they hope to address this later without making a point in the manuscript. This happens with multiple comments and issues, e.g., the point about a difference in HIPed material to material that has not been HIPed.

5) Why does it matter if other materials have an entirely different microstructure with, e.g., 70% gamma prime, if they perform better? A materials performance comparison is still valid!

6) Page 14 Ln 261: "The plot in Fig. 5 compares the high temperature properties of both ODS NiCoCr and NiCoCr with Re and B additions (ODS-ReB) (blue), GRX-810 (gold), and conventional wrought superalloys used commonly in AM (red)." – details of ODS-ReB are not added in Fig. 5.

7) Page 15 Ln 273: "One contributing factor is that non-ODS GRX-810 has higher strength than even previous ODS alloys, thus the creep stress is a lower proportion of the flow stress." – what are the authors trying to say?

8) Relative density measurements are still missing.

9) Extended Data Fig. 5 is shown in Fig. 4a,b – why show it again in the SI?

10) Unit typos in some places need to be corrected, e.g., Mpa to MPa.

It should be noted that simply saying that changes were made to the manuscript without even highlighting them in the manuscript makes it extremely hard for a reviewer to find and associate changes with the authors' responses (!).

Author Rebuttals to First Revision:

Dear Editors and Reviewers,

We are very excited to have been given the opportunity to revise our manuscript, “A 3D Printable Alloy Designed for Extreme Environments” for Nature a second time. We have carefully considered all the comments and concerns made by Reviewer #3, and in the list below we describe how each remark was addressed. Original comments and suggestions of Reviewer #3 are shown using **bold italics font**, while our responses are in regular font. Sections which were modified or newly incorporated into the manuscript body are highlighted in yellow. We are also greatly appreciative of the effort and time it took the Reviewers to provide their helpful insights. In addition, according to requests from Editors, we have made significant formatting changes to the paper to bring it within the guidelines for publication in Nature. These changes are described in detail separately below our responses to Reviewer #3.

Reviewer #3:

1) The authors include a new Extended Data Fig. 7 analysis but do not incorporate this into the discussion of the mechanisms around the performance of the material – which should be the main contribution of the work to the community that warrants publication in Nature (!). The formation of stress induced nitrides, for example, is mentioned in the figure caption but not discussed in the manuscript thereby retaining a sense of ‘report character’ of their work. Reference to the formation of such nitrides that are common in creep testing of such alloys are missing entirely as details about experimental characterization of their formation are missing; e.g., did they use EDS or any other technique?

- We greatly appreciate this important comment from the reviewer. After thoughtful consideration and taking into account length and format limitations as requested by the Editors, much more in-depth analysis and discussion related to the characterization of the differences in creep failures between GRX-810 and ODS-ReB was incorporated. We strongly feel that it has improved the overall impact of the manuscript. The below paragraph is now included at the end of the manuscript:

“While other ODS alloys (e.g. ODS-ReB) failed from a combination of grain boundary creep void coalescence and shear failure, GRX-810 appears to have suppressed these failure mechanisms as no apparent grain boundary voids/defects were observed after much longer test times. One contributing factor is that non-ODS GRX-810 has higher strength than even previous ODS alloys, thus the creep stress is a lower proportion of the alloy’s yield stress. Still, the grain boundary failure modes in other ODS alloys suggest that in GRX-810 the stable MC carbides and solute segregation of W, Cr, and Re along grain boundaries are significant factors contributing to the protection of the alloy from grain boundary failure mechanisms. Previous studies have hypothesized that carbide stability at high temperature will influence grain boundary crack initiation during creep⁵⁸. In addition, grain boundary diffusivity was reported to be correlated to the rate of void formation during creep^{59,60}. Therefore, the addition of W, Re (known slow diffusors) should further inhibit creep void formation along grain boundaries while Cr segregation is expected to improve grain boundary corrosion and oxidation properties⁶¹. Stress induced nitride formation was also observed in both the ODS-ReB (Cr-rich Nitrides) and GRX-810 (Al-rich nitride

and Cr-rich nitride) alloys. While the formation of these internal nitrides is considered detrimental to both alloys properties⁶², the nitrides in GRX-810 did not appear to contribute to grain boundary failure as in the ODS-ReB alloy.”

- We do not believe the nitrides played any notable role in better creep properties of GRX-810. Instead, all presented evidence correlated with literature suggests that the stable MC carbides and solute segregation of W, Re and Cr at the grain boundaries are indeed the most significant contributing factors to the superior creep properties of GRX-810 over the other ODS alloys.

- We apologize for omitting experimental details related to the nitride characterization. The statement below was included in the methods section highlighting that EDS was used to characterize the Nitrides.

“Chemical maps in SEM were obtained using an Oxford Ultim Max EDS Silicon Drift Detector operated by Aztec Software and were used to determine phases in post-crept samples.”

2) Extended Data Fig. 7b,c shows that Cr-nitrides initiate creep void formation eventually leading to failure. However, such stress induced nitride formation was suppressed in the GRX-810 alloys (Extended Data Fig. 7d,e) even after 2800 h under the same stress experienced by ODS-ReB (Extended Data Fig. 7b,c, which failed at 9 h). This could be because of other alloying element playing a role in suppressing such nitride formation in GRX-810 leading to the superior creep performance. What’s the authors take on this? Again, this might be essential to the manuscript and is yet not discussed.

- We thank the reviewer for the comment. We believe that this concern is now addressed by our response to comment (1).

3) Not being able to provide details about printing the material is an issues but we understand the importance of government control restrictions. Testing material in only one orientation or only conducting a single test in the second orientation is an issue and somewhat sloppy though - particularly as it doesn’t allow for a fair and statistically relevant comparison. Saying that this is common in the AM community may be true yet a problem with that community that could be ‘fixed’ by providing a complete set of data published in a high-profile journal like Nature. Saying that 1700 lbs of material have been received and results from the tests will be published later are weak arguments.

- We thank the reviewer for the comment. It is important to consider that to repeat the study exploring one or more different orientations would take over a year to complete with present resources, and therefore it is not feasible to accomplish for the present manuscript. In addition, in our experience (reading, reviewing, and producing manuscripts) providing a full dataset for multiple orientations has never been a requirement for publishing to even high impact journals (e.g., Martin *et al.* Nature 2017). Regarding consistency of the results, we understand the importance of making more statistical comparisons. Therefore, we have performed additional tensile tests at 1093°C on HIP GRX-810, As-built GRX-810, and non-ODS GRX-810. Using this new data, we have added standard deviation error bars to the micrograph in Fig. 3(b). The data shows how little spread in the tensile strength was observed for all three materials. We feel this adds further confidence to our conclusions.

Fig. 3: Mechanical tests of NiCoCr-based alloys. (a) Engineering stress-strain curves at 1093 °C for as-built and HIP alloys. (b) Comparison of ultimate tensile strength between different alloys. Wrought 718 and 625 strengths were provided by literature.⁴⁶ (c) 1093°C creep curves for as-built and HIP NiCoCr, NiCoCr-ODS, ODS-ReB at 20MPa. (d) The same tests with GRX-810 curves included. Additional tests of AM 718, 625, and H230 are shown at 20 MPa for a better comparison to conventional high temperature superalloys. **Error bars correspond to 1 standard deviation.**

4) It is somewhat discouraging that the authors state multiple times that they do not understand a certain materials performance, e.g., the decrease in ductility above 871C, and only mention in the response letter that they hope to address this later without making a point in the manuscript. This happens with multiple comments and issues, e.g., the point about a difference in HIPed material to material that has not been HIPed.

- We appreciate reviewer's remark. However, these comments are directed at aspects of materials performance that are clearly not the focus of this paper. It must be acknowledged that the present manuscript is primarily discussing the massive improvement in the mechanical properties of GRX-810 at 1,093°C (2,000°F). It should be considered that it is far beyond the scale and the scope of a single manuscript to expect that all aspects and observations be fully explored and characterized. Moreover, the analysis which is being suggested would not improve the impact of the paper or strengthen the main conclusions in any way.

5) Why does it matter if other materials have an entirely different microstructure with, e.g., 70% gamma prime, if they perform better? A materials performance comparison is still valid!

- We thank the reviewer for the comment. We decided to compare GRX-810 with polycrystalline, solid solution alloys for which data exists in the same test regime. Another main point of the manuscript is that GRX-810 can be 3D printed and we want to focus on comparing its properties to other alloys that have been shown to be readily 3D printable. Limited success of 3D printed single crystal alloys has been only rarely demonstrated (e.g. Korner/Eggeler paper) and requires a different processing route (selective electron beam melting). Thus, the application regime where GRX-810 has potential to be used is distinctly different from that for single crystal blade alloys. However, we have revised the figure title of Fig. 5 to “**Fig. 5: Creep rupture life of as-built GRX-810 compared to current SOA AM superalloys^{46,53-57}**” to make more clear what’s being compared for the readers. In addition, the title in the graph itself has also been changed to “**Creep Rupture Life of AM Alloys – 1,093°C**”.

6) Page 14 Ln 261: “The plot in Fig. 5 compares the high temperature properties of both ODS NiCoCr and NiCoCr with Re and B additions (ODS-ReB) (blue), GRX-810 (gold), and conventional wrought superalloys used commonly in AM (red).” - details of ODS-ReB are not added in Fig. 5.

- We are thankful to the reviewer for this important remark. To address this, we have separated the data between ReB-ODS and NiCoCr-ODS in Figure 5 as is shown below:

Fig. 5: Creep rupture life of as-built GRX-810 compared to current SOA AM superalloys^{46,53-57}. Scatter plot of superalloy creep rupture life at 1093°C. GRX-810 presents superior creep properties compared to wrought alloys currently used in 3D printed high temperature applications.

7) Page 15 Ln 273: “One contributing factor is that non-ODS GRX-810 has higher strength than even previous ODS alloys, thus the creep stress is a lower proportion of the flow stress.” - what are the authors trying to say?

- We thank the reviewer for the comment. We have modified the statement for clarity:

- “One contributing factor is that non-ODS GRX-810 has higher strength than even previous ODS alloys, thus the creep stress is a lower proportion of the alloy’s yield stress.”

8) Relative density measurements are still missing.

- We sincerely apologize for omission of relative density measurements description. We have now addressed the issue by including the relative density measurements in the methods section as follows:

- “The Archimedes method with deionized water as the immersion fluid was used to determine the density of the additively manufactured GRX-810 material. Specimens that have surface-breaching networks of cracks and porosity allow water infiltration which can be observed as bubbling during submersion. No bubbling was observed during the submersion of the GRX-810. Measurements of the part's mass in air (M_a) and mass in water (M_w) were taken on a Mettler Toledo XS205 system. Density of the AM part was calculated by the equation below.

$$p = \left(\frac{M_a}{(M_a - M_w)} \right) \times (p_w - p_0) + p_0 \quad (1)$$

Where p_w is the temperature-dependent density of water and p_0 is the air density. The reported density value is an average of three independent measurements.

Powder feedstock (15 μm – 45 μm) of the optimized composition revealed in Fig. 1 was obtained, coated with Y_2O_3 nanoparticles, and built with L-PBF using the steps provided in the methods section. Successful production of GRX-810 using AM L-PBF allowed for the characterization of GRX-810 in both the as-built and post hot isostatic pressed (HIP) states. Extended Data Fig. 6 reveals the high density (>99.97 %) that can be achieved with optimized print parameters for GRX-810 based on optical microscopy analysis. **Relative density measurements further confirmed this value revealing 99.96% density value for the same sample.”**

9) Extended Data Fig. 5 is shown in Fig. 4a,b - why show it again in the SI?

- We thank the reviewer for the comment. We would like to clarify that the extended data figure shows the test data all the way out to 100 hours. However, in Fig. 4 only up to 35 hours are shown due to limited space being available.

10) Unit typos in some places need to be corrected, e.g., Mpa to MPa.

- We sincerely apologize for these errors and typos. We have carefully checked the text and all cases of Mpa have been changed to MPa. Other errors and typos were corrected as well.

It should be noted that simply saying that changes were made to the manuscript without even highlighting them in the manuscript makes it extremely hard for a reviewer to find and associate changes with the authors' responses (!).

- We thank the reviewer for the comment. However, according to journal standards, we have included a draft with highlighted tracked changes when we resubmitted the manuscript. We have done the same with this revision. We hope that the reviewers have access to that copy which has changes tracked and highlighted. If that is not a case, we believe this issue must be resolved in direct communication between the Reviewers and the Editors.

Editor:

LENGTH: We estimate the current length of your paper to be ~3500 words, which exceeds our usual limit by a considerable margin. With 5 display items as at present, the main text of the revised version should be no more than ~2800 words. Keep in mind that important technical details that are not central to the main message of the paper can be moved into the Methods section or, if necessary, a Supplementary Information section (see below).

- We thank the editor for the comment. We have now reduced the overall word count of the main manuscript from 3500 words to 2700 words.

TITLES: Titles cannot exceed 75 characters (including spaces); they must not contain punctuation. I suggest you remove the name of the alloy from the title to make the title more approachable.

- We appreciate this remark. To address it, "GRX-810:" was removed from the title.

SUMMARY PARAGRAPH: All Nature papers begin with a fully referenced paragraph, typically no longer than 200 words, aimed at readers in other disciplines. This paragraph starts with a 2- to 3-sentence, basic introduction to the field; continues with a 1-sentence statement of the main findings starting 'Here we show' or an equivalent phrase; and finally, concludes with 2 to 3 sentences putting the main findings into general context so it is clear how the results described in the paper have moved the field forward. A downloadable, annotated example is available at <https://www.nature.com/nature/for-authors/formatting-guide>.

Please reference your summary paragraph.

- We appreciate the comment. The summary paragraph is now fully cited and matches the structure specified above.

MAIN TEXT: Further introductory material in the main text of the paper should not be necessary. Any discussion at the end of the paper should also be brief, and not repeat what is already written in the initial summary paragraph. Subheadings of up to 40 characters (including spaces) may be used to separate sections of the paper if this aids navigation. You are only allowed 1 level of subheadings, so I would suggest removing 'Results' and 'Discussion' and keep your more focused headings, making sure they do not exceed 40 characters.

- We thank the editors for the suggestion. All subheadings are now under 40 characters.

REFERENCES: As a guideline, most papers should need no more than 50 references in the main text; additional references can be cited in (and listed after) the Methods section, as detailed above. Please move 20 references to the Additional References section, which will appear after the Methods.

- We are thankful to the editor for this suggestion. The additional references have now been moved after the methods section.

FIGURE FORMATTING: Lettering in all figures (labelling of axes and so on) should be in uniform, sans-serif font, in lower-case type, and large enough to permit substantial reduction for publication (minimum font size 5 pt). Separate parts of a figure are labelled a, b, etc. Units have a single space between the number and the unit, and follow SI nomenclature or the nomenclature common to a particular field. Thousands are separated by commas (1,000). Unusual units or abbreviations are defined in the legend. Scale bars rather than magnification factors should be used.

We thank for this remark. Thousands are now separated by commas throughout the manuscript.

Again, we hope these changes and additions incorporated into the manuscript clearly show how serious we considered the concerns from reviewer 3. We feel that this new manuscript is stronger and more successfully presents our findings. We would like to thank all three reviewers and the editors for their time and constructive feedback which we feel has significantly improved the quality of the paper.

Regards,

Tim Smith